# NADH-bound AIF activates the mitochondrial CHCHD4/MIA40 chaperone by a substrate-mimicry mechanism

Chris A Brosey (iD)[1], Runze Shen (iD)[1] & John A Tainer (iD)[1,2,3✉]

## Abstract

**Mitochondrial metabolism requires the chaperoned import of disulfide-stabilized proteins via CHCHD4/MIA40 and its enigmatic interaction with oxidoreductase Apoptosis-inducing factor (AIF). By crystallizing human CHCHD4's AIF-interaction domain with an activated AIF dimer, we uncover how NADH allosterically configures AIF to anchor CHCHD4's β-hairpin and histidine-helix motifs to the inner mitochondrial membrane. The structure further reveals a similarity between the AIF-interaction domain and recognition sequences of CHCHD4 substrates. NMR and X-ray scattering (SAXS) solution measurements, mutational analyses, and biochemistry show that the substrate-mimicking AIF-interaction domain shields CHCHD4's redox-sensitive active site. Disrupting this shield critically activates CHCHD4 substrate affinity and chaperone activity. Regulatory-domain sequestration by NADH-activated AIF directly stimulates chaperone binding and folding, revealing how AIF mediates CHCHD4 mitochondrial import. These results establish AIF as an integral component of the metazoan disulfide relay and point to NADH-activated dimeric AIF as an organizational import center for CHCHD4 and its substrates. Importantly, AIF regulation of CHCHD4 directly links AIF's cellular NAD(H) sensing to CHCHD4 chaperone function, suggesting a mechanism to balance tissue-specific oxidative phosphorylation (OXPHOS) capacity with NADH availability.**

**Keywords** Apoptosis-inducing Factor (AIF); CHCHD4/MIA40; OXPHOS; X-ray Crystallography; Small-angle X-ray Scattering (SAXS)
**Subject Categories** Membranes & Trafficking; Organelles; Structural Biology

## Introduction

Mitochondria rely upon essential protein transport pathways to import and assemble 99% of their proteome from nuclear-encoded precursors (Rath et al, 2021). As gatekeepers of the mitochondrial proteome, these import assemblies serve as sensors of the cellular environment to deliver quality-controlled proteins to mitochondria in response to metabolic demand and cellular stress (Harbauer et al, 2014). As one of five major protein import networks (Wiedemann and Pfanner, 2017), the CHCHD4/MIA40 disulfide relay pathway provides coupled import and chaperoned folding of the CHCH (coiled-coil-helix-coiled-coil-helix) family of proteins, which contain disulfide-stabilized helical hairpins identified by their signature twin $CX_3C$ and $CX_9C$ motifs (Al-Habib and Ashcroft, 2021; Mordas and Tokatlidis, 2015; Reinhardt et al, 2020). Notably, this unique class of proteins assemble and participate in critical mitochondrial machines, including respiratory Complexes I and IV, the cristae-remodeling MICOS complex, and mitochondrial import assemblies (Erdogan and Riemer, 2017; Habich et al, 2019).

The mitochondrial disulfide relay is mediated within the mitochondrial intermembrane space (IMS) by the CHCHD4/MIA40 chaperone, itself a member of the CHCH family (Al-Habib and Ashcroft, 2021; Finger and Riemer, 2020; Mordas and Tokatlidis, 2015). The chaperone engages nascent polypeptide substrates through a hydrophobic docking motif within the substrate's C-terminal hairpin helix (Banci et al, 2010; Sideris et al, 2009). Binding to CHCHD4 induces helical folding and temporarily links the hairpin's third cysteine to CHCHD4's redox-active cysteine–proline–cysteine (CPC) motif (Banci et al, 2010; Banci et al, 2009; Sideris et al, 2009). Shuffling the mixed disulfide back to the substrate creates the hairpin's interior disulfide at the polypeptide turn, facilitating energetically favorable formation of the exterior disulfide at the end of the hairpin. This stabilizes the hairpin fold and traps the substrate within the IMS to complete protein import (Banci et al, 2010; Habich et al, 2019; Sideris et al, 2009). CHCHD4's reduced CPC motif is regenerated for subsequent import cycles by FAD-dependent oxidoreductase Augmenter-of-Liver-Regeneration (ALR) (Banci et al, 2011), which utilizes cytochrome C and ultimately respiratory Complex IV as final electron acceptors of the relay (Allen et al, 2005; Peker et al, 2021; Tang et al, 2020). Because of its gatekeeper role for essential mitochondrial proteins and processes, disruption of CHCHD4 (Hangen et al, 2015; Sokol et al, 2018), ALR (Calderwood et al, 2016; Di Fonzo et al, 2009; Nambot et al, 2017), or interactions with

[1]Department of Molecular and Cellular Oncology, The University of Texas M. D. Anderson Cancer Center, Houston, TX 77030, USA. [2]Department of Cancer Biology, The University of Texas M. D. Anderson Cancer Center, Houston, TX 77030, USA. [3]MBIB Division, Lawrence Berkeley National Laboratory, Berkeley, CA 94720, USA.
✉E-mail: JTainer@mdanderson.org

hairpin substrates (Erdogan and Riemer, 2017; Lehmer et al, 2018; Modjtahedi et al, 2016; Wang et al, 2021; Zhou et al, 2017) results in mitochondrial dysfunction and disease.

The disulfide relay pathway is highly conserved in eukaryotes. Fungal, plant, and animal CHCHD4/MIA40 share a hydrophobic, redox-active CHCH central domain and a flexible, positively charged C-terminus that stabilizes CHCHD4's cytosolic precursor (Banci et al, 2009; Murschall et al, 2020) (Fig. 1A). The N-terminal region of the chaperone, however, fundamentally differs between fungi (where the chaperone is known as Mia40) and higher eukaryotes. In *S. cerevisiae*, a well-studied model system for mitochondrial import, an extended N-terminus tethers Mia40 to the inner mitochondrial membrane (IMM), allowing the central catalytic domain to flexibly extend into the IMS (Mordas and Tokatlidis, 2015). Among more complex eukaryotes, however, the CHCHD4 N-terminus lacks this mitochondria-targeting membrane tether and instead forms an NAD(H)-regulated complex with IMM-anchored oxidoreductase Apoptosis-Inducing Factor (AIF) (Hangen et al, 2015; Salscheider et al, 2022). NADH specifically activates AIF to bind CHCHD4 (Hangen et al, 2015; Salscheider et al, 2022), but the molecular mechanism underlying their association remains unknown. NADH is known to allosterically stimulate AIF dimerization and release a 50-residue disordered loop from AIF's surface by reducing and forming a stable charge-transfer complex (CTC) with AIF's FAD cofactor (Brosey et al, 2016; Churbanova and Sevrioukova, 2008; Sevrioukova, 2009). These changes in AIF architecture are presumed to enable binding to CHCHD4; yet, how they specifically configure the binding interface and enable functional binding remains unclear.

Besides physical coupling of CHCHD4 to the IMM, AIF supports mitochondrial proteostasis of mammalian CHCHD4 (Hangen et al, 2015) and stimulates CHCHD4-dependent import of select substrates essential to Complex I (Salscheider et al, 2022). AIF knockdown in certain human cell lines results in diminished CHCHD4 protein levels (Hangen et al, 2015; Meyer et al, 2015), while mouse models targeting AIF expression or disease-causing mutations reveal decreased levels of CHCHD4 substrates required for respiratory complex biogenesis and oxidative phosphorylation (OXPHOS) (Hangen et al, 2015; Vahsen et al, 2004; Wischhof et al, 2018). Similarly, many human patients with disease-related AIF mutations also exhibit reduced OXPHOS protein levels and function (Ardissone et al, 2015; Berger et al, 2011; Ghezzi et al, 2010; Hu et al, 2017; Kettwig et al, 2015; Moss et al, 2021; Rinaldi et al, 2012). However, how NADH-activated AIF functionally supports CHCHD4 persistence and import activity is not fully understood. Past studies have pointed to a role for AIF in co-translational import of CHCHD4 into mitochondria (Hangen et al, 2015). Whether AIF constitutes a novel import chaperone of CHCHD4 or simply enables the existing disulfide relay to import and activate nascent CHCHD4 polypeptides has remained enigmatic.

Here, we have uncovered how NADH activation of AIF regulates the AIF-CHCHD4 complex by crystallizing CHCHD4's N-terminal interaction domain complexed to an activated AIF dimer and solving the detailed structure. The high-resolution crystal structure reveals a conserved CHCHD4 β-hairpin/helix motif which hydrophobically scaffolds AIF's C-terminal domain, occupying a surface allosterically exposed by release of AIF's C-loop. While targeted mutation of this novel AIF-interaction

motif disrupts CHCHD4 association with endogenous AIF in cells, it does not hinder CHCHD4 mitochondrial localization, arguing against AIF as a novel chaperone of CHCHD4 import. Significantly, CHCHD4's AIF-interaction motif mimics the hydrophobic docking signatures of CHCHD4's native import substrates. By employing NMR and SAXS, we demonstrate that the AIF-interaction motif engages and shields the redox-sensitive active site, pointing to intramolecular substrate mimicry as a novel regulator of CHCHD4 activity. Consistent with this, selectively disrupting the AIF-interaction domain increases CHCHD4 affinity and activity toward unfolded $CX_9C$ substrates. We further establish that binding between this substrate-mimicking regulatory domain and NADH-activated AIF enhances wild-type CHCHD4 substrate affinity and stimulates its disulfide chaperone activity. Collectively, these results reveal AIF as a mitochondrial NAD(H) sensor and CHCHD4 effector and uncover a novel role for CHCHD4's AIF-interaction domain in protecting, regulating, and activating the CHCHD4 active site.

# Results

## The AIF-CHCHD4 complex is enabled by NADH-stimulated release of AIF's C-loop

Redox binding to NADH and subsequent CTC (NAD$^+$/FADH$^-$) formation activates AIF association with CHCHD4 (Hangen et al, 2015). This implicates NADH-stimulated allosteric dimerization of AIF and/or release of AIF's surface C-loop in configuring the CHCHD4 binding surface (Appendix Fig. S1A). To probe links between AIF allostery and CHCHD4 association, we established a microscale thermophoresis (MST) binding assay to screen previously characterized AIF allostery mutants (Brosey et al, 2016) for their ability to engage CHCHD4 (Fig. 1; Appendix Fig. S1C). Supporting and extending previous reports (Hangen et al, 2015), binding between wild-type AIF and CHCHD4 is centered within CHCHD4's N-terminal interaction domain and does not require a redox-active CPC motif (C53S/C55S or SPS) (Fig. 1B). Among metazoans, this N-terminal interaction domain (N45) is strikingly conserved relative to CHCHD4's variable C-terminal region (CT), suggesting a conserved functional engagement with AIF (Appendix Fig. S1B).

The selected AIF allostery mutants dimerize without NADH ligand (Brosey et al, 2016). We tested two classes of dimer-permissive mutants, containing either dissociated or bound C-loops. Dimer-permissive AIF mutants with constitutively released or absent C-loops (W196A, R529A, ΔC-loop) can bind CHCHD4 without NADH (Fig. 1C,D; Table 1). In contrast, dimeric AIF mutants with intact, surface-bound C-loops (H454A, S480A) bind weakly or not at all. Thus, displacement of AIF's C-loop from the protein surface is an important allosteric prerequisite for enabling the AIF-CHCHD4 complex.

## CHCHD4 hydrophobically scaffolds AIF's allosterically exposed C-terminal domain

To understand how C-loop displacement stimulates binding between AIF and CHCHD4, we sought to crystallize their interaction complex. Size-exclusion chromatography coupled to multi-angle light scattering (SEC-MALS) of AIF-CHCHD4

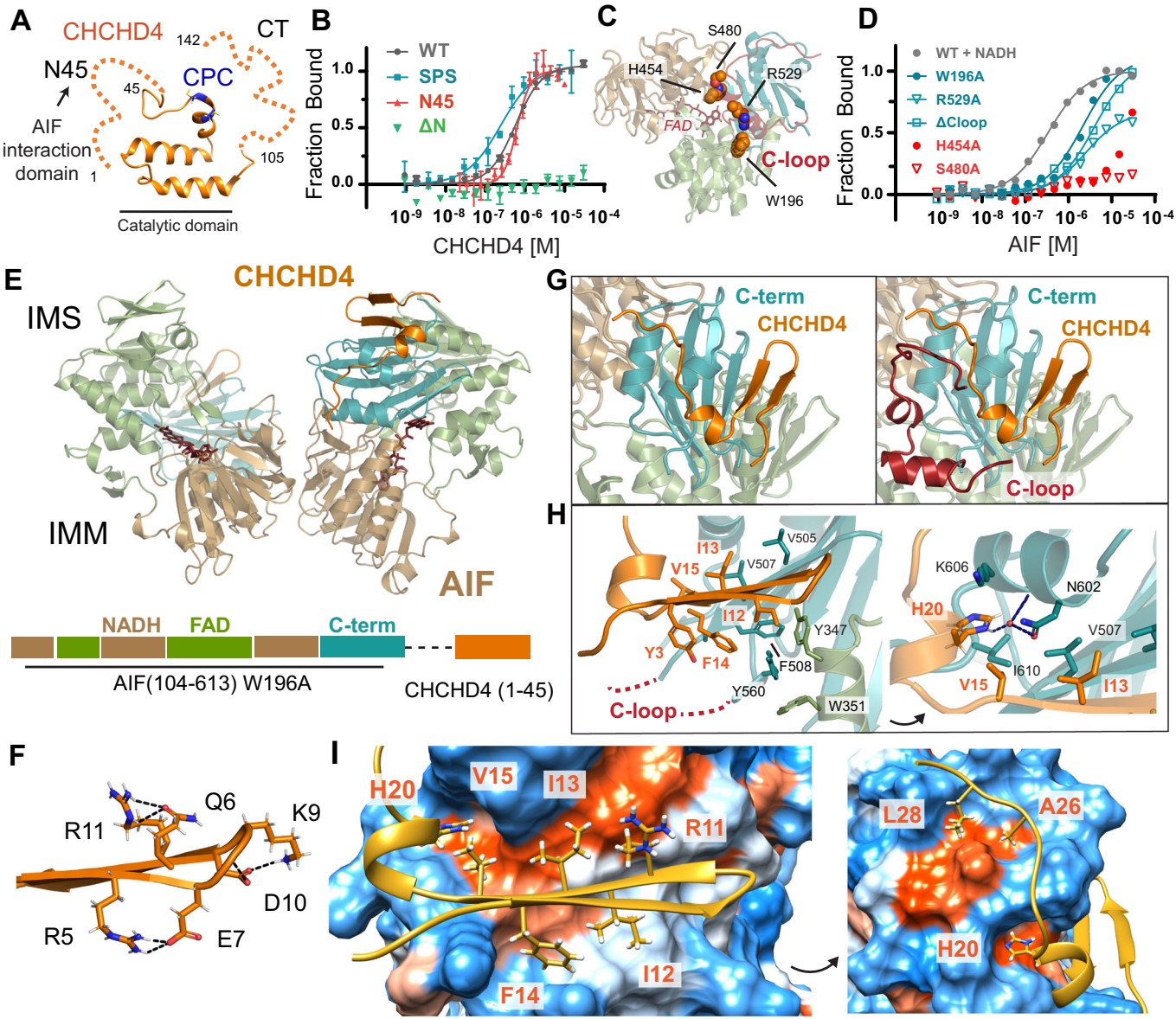

**Figure 1. NADH-stimulated allosteric displacement of AIF's C-loop allows CHCHD4 to hydrophobically scaffold AIF's C-terminal domain.**

(**A**) CHCHD4's central CHCH domain with catalytic CPC motif is flanked by an N-terminal AIF-interaction domain (residues 1–45, N45) and a flexible C-terminus that stabilizes CHCHD4's cytosolic precursor (residues 105–142, CT). Coordinates of the catalytic domain are from PDB entry 2K3J. (**B**) Wild-type CHCHD4, a catalytically inactive mutant (C53S/C55S–SPS), and the isolated N-terminal AIF-interaction domain (N45) bind NADH-activated AIF, while an N-terminal truncation mutant (ΔN, residues 40–142) does not. Each MST curve represents the average of three thermophoresis scans of a representative binding titration into Atto488-AIF(78–613)/NADH; error bars are standard deviations. Binding titrations were repeated at least three times. See also Table 1. (**C**) AIF residues W196, H454, S480, and R529 (spheres) allosterically regulate AIF dimerization and C-loop release. Coordinates of monomeric AIF are from PDB entry 4BV6 (**D**) CHCHD4 preferably engages allosteric AIF mutants with open C-loops in the absence of NADH. Each curve represents the average of three thermophoresis scans of a representative binding titration into Atto488-CHCHD4. Binding titrations were repeated two to three times. See also Table 1. (**E**) CHCHD4 scaffolds AIF's C-terminal domain in the crystallized complex. (**F**) CHCHD4's interaction domain forms a β-hairpin flanked by internal salt bridges. (**G**) Allosteric release of AIF's C-loop exposes the CHCHD4 binding site. The AIF-CHCHD4 crystal structure is superimposed with the bound C-loop from PDB: 4BV6. (**H**) C-loop boundary residues (F508, Y560) align to form an aromatic tetrad with Y347 and W351 at AIF's interface with the CHCHD4 β-hairpin (left), while CHCHD4's anchoring histidine (H20) inserts into a hydrophobic pocket formed by I610 and V507. Black dotted lines indicate hydrogen bond contacts. (**I**) A Kyle–Doolittle surface rendering of the AIF binding surface highlights the hydrophobic (orange) interface with CHCHD4. Source data are available online for this figure.

complexes formed with excess NADH indicates a mixture of one or two CHCHD4 molecules bound to a single AIF dimer and upholds dimeric AIF as a relevant binding target for CHCHD4 (Appendix Fig. S1C–E; Table EV1). The polydispersity of the complex, however, indicates sensitivity to dilution-induced dissociation.

To generate a stable complex for crystallization, we engineered a custom construct linking CHCHD4's N-terminal interaction domain (N45) to the dimer-permissive AIF (104–613) W196A mutant (Brosey et al, 2016) which maintains an open C-loop (Fig. 1E). We verified that this chimeric construct mimics AIF-CTC

**Table 1.** MST affinity measurements between AIF and CHCHD4 mutants.

| CHCHD4 mutants (target Atto488-AIF-NADH, Fig. 1) | | | | |
|---|---|---|---|---|
| | **K$_d$ (µM)** | **95% CI** | **Goodness-of-fit (R$^2$)** | |
| **CHCHD4-WT** | **0.58** | **0.49–0.68** | **0.995** | |
| SPS | 0.23 | 0.18–0.29 | 0.990 | |
| N45 | 0.63 | 0.55–0.73 | 0.986 | |
| ΔN | ND | ND | ND | |

| AIF allosteric mutants (target Atto488-CHCHD4, Fig. 1) | | | | |
|---|---|---|---|---|
| | **K$_d$ (µM)** | **95% CI** | **Goodness-of-fit (R$^2$)** | |
| **AIF + NADH** | **0.34** | **0.30–0.38** | **0.997** | |
| W196A | 2.2 | 1.6–2.9 | 0.9863 | |
| R529A | 2.5 | 1.9–3.2 | 0.9891 | |
| ΔCloop | 4.0 | 3.1–5.3 | 0.9893 | |
| H454A | ND | ND | ND | |
| S480A | ND | ND | ND | |

| CHCHD4 mutants (target Atto488-AIF-NADH, Fig. 3) | | | | |
|---|---|---|---|---|
| | **K$_d$ (µM)** | **95% CI** | **Hill coefficient** | **95% CI** | **Goodness-of-fit (R$^2$)** |
| **CHCHD4-WT** | **0.18** | **0.16–0.21** | **1.6** | **1.3– 2.1** | **0.9802** |
| AIA | 3.0 | 1.6–5.7 | 1 | N/A | 0.9158 |
| AIA-A | ND | ND | ND | ND | ND |
| H20D | ND | ND | ND | ND | ND |
| L28D | 0.24 | 0.18–0.33 | 1.75 | 1.1–3.2 | 0.9673 |

Wild-type reference measurements are highlighted in bold. Confidence intervals (CI) at 95% represent the standard error of fitting from three averaged MST scans of a representative binding titration.

architecture via limited proteolysis (C-loop dissociation), in vitro cross-linking (dimerization), and small-angle X-ray scattering (SAXS) to define global architecture in solution (Appendix Fig. S2A,B and Appendix Table S1). While full-length CHCHD4 engages AIF-W196A, it does not bind the AIF-W196A-CHCHD4-N45 chimera, indicating that the linked N45 fragment occupies the CHCHD4 binding site (Appendix Fig. S2C).

The AIF-CHCHD4 chimera crystallizes in conditions resembling the original AIF-W196A mutant (Brosey et al, 2016), allowing the structure to be solved by molecular replacement (Appendix Table S2). The structure reveals the expected conserved AIF dimer (Fig. 1E). The CHCHD4 N-terminal domain is resolved for both AIF monomers, allowing the first 30 (Molecule 1) and 20 (Molecule 2) residues, respectively, to be modeled into the electron density (Appendix Fig. S2D). B-factors for the primary CHCHD4 N-terminal domain are consistent with those of both AIF monomers (average B-factor 51.91, Appendix Table S3), while the B-factor average of the second molecule runs higher (average B-factor 73.67). The structure reveals that the dissociation of AIF's C-loop exposes a unique binding surface, allowing CHCHD4 to form a hydrophobic scaffold around AIF's C-terminal domain (Fig. 1E,G).

The CHCHD4 N-terminus assumes a unique β-hairpin/helix motif when complexed with AIF with a buried solvent accessible surface area of 1440 Å. The first twenty residues form a β-hairpin, which pairs with the core β-sheet of AIF's C-terminal domain in a parallel orientation (Fig. 1F,G). The interface with AIF is mediated by hydrophobic side chains of β-hairpin residues I12, I13, F14, and V15 which complement the exposed hydrophobic edge of AIF's C-terminal domain (Fig. 1H,I). From the β-hairpin, the CHCHD4 polypeptide chain makes a helical turn and wraps onto the surface exposed by the C-loop. The helical turn is anchored by a histidine (H20) which inserts into a hydrophobic pocket formed by AIF residues I610, V507, V564, and A604 and makes a water-mediated contact with N602 (Fig. 1H, right panel). CHCHD4 residues A26 and L28 anchor the peptide onto the surface of AIF's C-terminal domain, targeting hydrophobic patches previously shielded by the C-loop (Fig. 1I, right panel).

C-loop boundary residues F508 and Y560 contribute to AIF's hydrophobic binding surface, forming an aromatic tetrad with Y347 and W351 (Fig. 1H, left panel) as previously observed in crystal structures of the activated AIF dimer (Ferreira et al, 2014; Sevrioukova, 2009). CHCHD4 packs against the tetrad, which secures and orients the C-loop away from the C-terminal domain. This arrangement maintains the C-loop's displaced state and may prevent its competition for the CHCHD4 binding surface.

To compare the crystal structure to the solution state observed by small-angle X-ray scattering, we added residues missing from the crystal structure and submitted the resulting model to BilboMD simulation and minimal ensemble search (MES). BilboMD generates a population of models via constrained molecular dynamics and identifies the minimum number of models that best describe the data. For the AIF-CHCHD4 chimera, the MES algorithm returned a two-model ensemble (Appendix Fig. S3A,B), supporting flexibility at the N- and C-termini and C-loop. Notably, one of the CHCHD4-N45 fragments of the minor population is

captured in an unbound state, suggesting an equilibrium between bound and unengaged CHCHD4-N45 in solution that favors the AIF-CHCHD4–N45 complex.

The overall hydrophobic character of the AIF-CHCHD4 interface supports the stable complex observed in vivo (Salscheider et al, 2022) and underscores AIF as an IMM-anchor for metazoan CHCHD4. Leveraging our X-ray structure and the published NMR structure of the human CHCHD4 catalytic domain (PDB: 2K3J) (Banci et al, 2009), we constructed a model of full-length CHCHD4 flexibly extending from the AIF-anchored N-terminal domain (Fig. EV1). The model indicates that CHCHD4 complexed with AIF would be oriented and elevated toward the outer mitochondrial membrane, well positioned to receive incoming polypeptide substrates. The model also reveals that the AIF dimer separates the redox-active CHCH domains of opposing CHCHD4 molecules, reducing the potential for mutual interference when handling unfolded substrates. Thus, a single AIF dimer could support two CHCHD4 import centers. Furthermore, pairing CHCHD4 chaperones in this manner may increase regeneration efficiency by allowing regeneration partner ALR, also a dimer, to cyclically exchange between the two chaperones.

## The AIF-interaction motif is conserved across metazoan CHCHD4 homologs

CHCHD4's AIF-interaction domain exhibits high sequence conservation among metazoans (Appendix Fig. S1B). To determine if the unique architecture of CHCHD4's AIF-interaction domain is preserved beyond humans, we analyzed structural models of metazoan CHCHD4 homologs from the AlphaFold Protein Structure Database (Jumper et al, 2021; Varadi et al, 2022). As a benchmark, we compared the predicted AlphaFold model of human CHCHD4 to experimental structures of the AIF-interaction domain and central CHCH catalytic domain (Banci et al, 2009) (Fig. EV2A). The AlphaFold model aligns remarkably well to both experimental structures [AIF-interaction domain—$C_\alpha$ RMSD 0.94 Å; catalytic CHCH domain—$C_\alpha$ RMSD 1.8 Å to first NMR conformer], supporting a comparative analysis of select CHCHD4/Mia40 homologs.

Examination of vertebrate CHCHD4 models reveals a shared hydrophobic AIF-interaction motif supported by the β-hairpin/helix arrangement observed in the human structure (Fig. EV2B,C; Appendix Fig. S4A). This similarity also extends to the invertebrate fruit fly *Drosophila*. In *C. elegans*, the β-hairpin is absent, but the hydrophobic AIF-interaction motif and helical element are retained by the N-terminus (Fig. EV2D). This contrasts with fungal Mia40 homologs retrieved from the AlphaFold database (Fig. EV2E; Appendix Fig. S4B). While their central catalytic domains exhibit high structural conservation across Mia40 models, there is high variation and low predictive confidence among the associated N-terminal regions. The absence of a systematic N-terminal organization among fungal homologs is consistent with a simpler regulatory environment for protein import and mitochondrial responses in these organisms.

## CHCHD4's AIF-interaction motif is not essential for mitochondrial localization

We probed the AIF-CHCHD4 interface by preparing CHCHD4 mutants and assessing binding to activated AIF-NADH dimers with MST. Mutating hydrophobic residues at the β-hairpin (I12A/F14A–AIA) weakened affinity for AIF. Targeting the histidine anchor through charge reversal (H20D) or combining hairpin and histidine mutants (I12A/F14A/H20A–AIA-A) abolished AIF-interaction (Fig. 2A; Table 1). The combination mutant (AIA-A) also failed to form a productive complex with the AIF-NADH dimer when assessed by SEC-MALS (Fig. 2B; Table EV1). In contrast, charge substitution for L28, which contacts the exposed C-loop surface, resulted in limited change to AIF affinity (Fig. 2A).

To assess binding in a cellular context, we screened CHCHD4 mutants for their ability to co-immunoprecipitate endogenous AIF from 293T cells (Fig. 2C). Wild-type CHCHD4 and the catalytically inactive SPS mutant successfully associated with AIF, but AIA (β-hairpin), AIA-A (β-hairpin/histidine), and N-terminally truncated mutants (ΔN) did not. The combined in vitro and cellular interaction data corroborate the CHCHD4 interface revealed by the crystal structure and identify the CHCHD4 β-hairpin and histidine anchor as critical contact points for engaging the activated AIF dimer.

AIF-CHCHD4 association supports CHCHD4 mitochondrial localization (Hangen et al, 2015). Using knowledge of the AIF-CHCHD4 interface, we tested a model for AIF as a direct import chaperone by monitoring the mitochondrial localization of CHCHD4 interaction mutants in U2-OS cells. Wild-type CHCHD4 exhibits an expected mitochondrial distribution (Fig. 2D). However, targeting CHCHD4's AIF-interaction motif (AIA-A) or removing the N-terminus (ΔN) does not prevent CHCHD4 mitochondrial localization. We also tested the N-terminal interaction domain (N45, residues 1–45) as a stand-alone mitochondrial localization signal by fusion to GFP, but this construct remained cytoplasmic (Fig. EV3A).

As preventing AIF-interaction has limited impact on CHCHD4 mitochondrial localization, we tested the effect of disrupting CHCHD4's interaction with the disulfide relay import system as a control. For this, we altered CHCHD4's own internal twin $CX_9C$ disulfide import motif (SX9S) to prevent productive engagement with native CHCHD4 chaperones. We also examined the catalytically inactive SPS mutant, which can function as a dominant-negative inhibitor of disulfide relay activity (Habich et al, 2019). The SX9S mutant exhibits diffuse cellular localization, indicating CHCHD4 exclusion from mitochondria. The SPS mutant displays heterogeneous localization, appearing mitochondrial in cells with lower mutant expression, but shifting to an increasingly cytoplasmic distribution for cells with higher mutant expression (Fig. 2D). Notably, both mutants successfully interact with endogenous AIF in whole-cell lysates, despite their variable trafficking to mitochondria (Fig. EV3B). Thus, AIF does not exercise a direct import function toward CHCHD4; however, productive contact with an intact disulfide relay is necessary to ensure successful CHCHD4 mitochondrial localization.

## The AIF-interaction motif shields CHCHD4's redox-active site by mimicking unfolded import substrates

Strikingly, CHCHD4's AIF-interaction motif shares sequence and surface similarities with hydrophobic docking signatures from several CHCHD4 import substrates (Banci et al, 2010; Sideris et al, 2009) (Appendix Fig. S5A). This resemblance suggests the potential for intramolecular association between CHCHD4's AIF-interaction

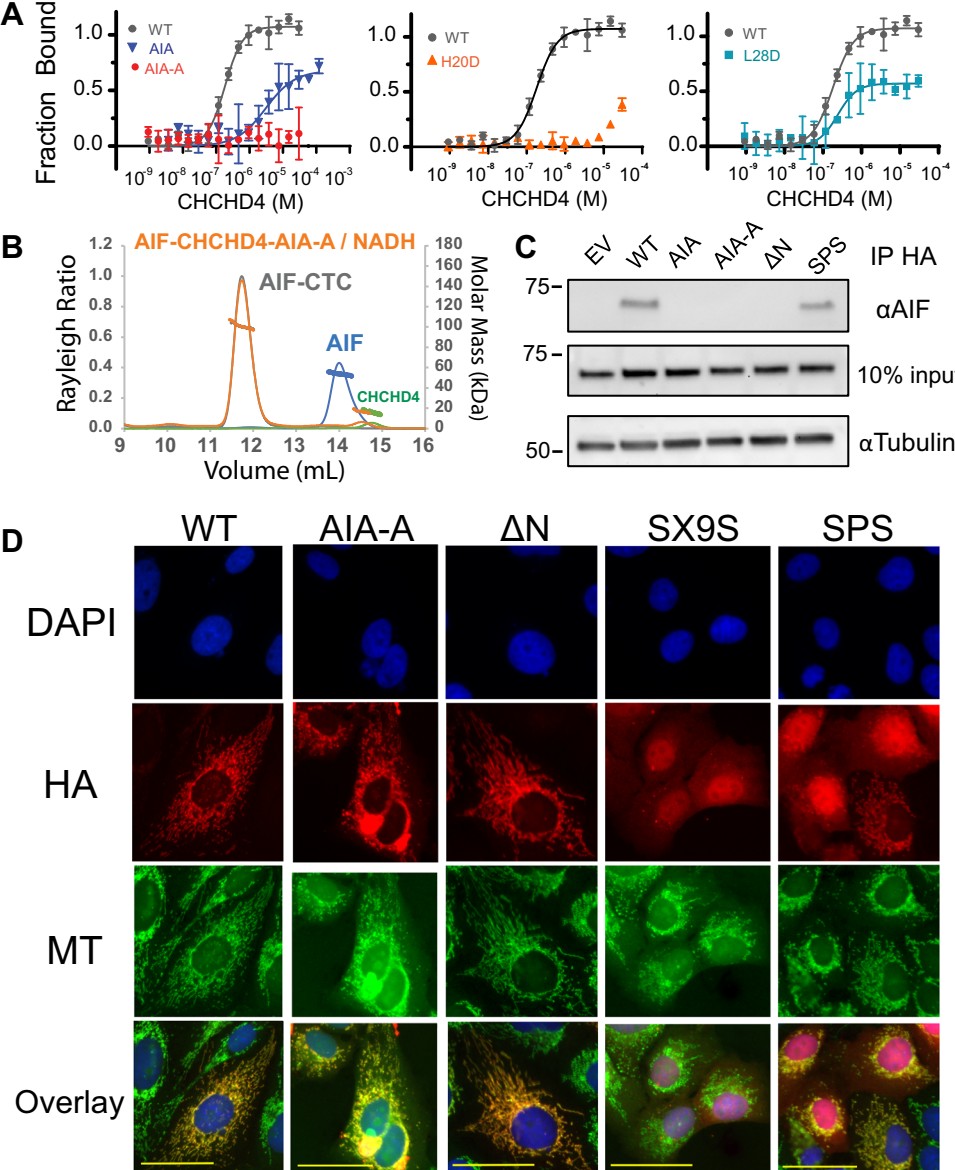

**Figure 2. CHCHD4's AIF-interaction motif enables association with endogenous AIF but is not required for mitochondrial localization.**

(A) Targeting the AIF-interaction motif inhibits binding between CHCHD4 and Atto488-AIF/NADH. Each MST curve represents the average of three thermophoresis scans of a representative binding titration into Atto488-AIF/NADH. Binding titrations were repeated at least three times. See also Table 1. (B) The CHCHD4 β-hairpin/helix combination mutant AIA-A fails to form a complex with AIF-CTC as monitored by SEC-MALS. See also Table EV1 and Appendix Fig. S1. Results are representative of two independent experiments (C) CHCHD4 mutants that target the AIF-interaction motif (AIA, AIA-A) or delete the interaction domain (ΔN) fail to immunoprecipitate endogenous AIF from 293T cells. Results are representative of three independent experiments. (D) Immunofluorescent localization of transiently expressed HA-tagged CHCHD4 mutants in U2-OS cells. Mutants defective for AIF-interaction (AIA-A, ΔN) exhibit mitochondrial localization, while mutants defective for disulfide relay import (SX9S, SPS) are mitochondrially excluded. Hoescht (DAPI), Mitotracker (MT). The scale bar is 50 microns. Results are representative of three independent experiments. Source data are available online for this figure.

domain and catalytic CHCH core. Aligned with this, the combination β-hairpin/histidine mutant (AIA-A), which targets hydrophobic substrate-mimicking residues I12 and F14, assumes an extended conformation when probed by SEC-MALS and SAXS, consistent with intramolecular disruption (Fig. 3A; Appendix Fig. S5B; Table EV1; Appendix Table S1).

To examine intramolecular contact between CHCHD4's AIF-interaction and catalytic domains directly, we collected

$^1$H-$^{15}$N-HSQC NMR spectra on $^{15}$N-enriched full-length and truncated (ΔN) CHCHD4 proteins engineered with inactive CPC motifs (SPS) to reduce disulfide interference (Fig. 3B). Amide backbone resonances measured by NMR are highly sensitive to local chemical environment. Thus, the existence of intramolecular contact within CHCHD4 should manifest as chemical shift perturbations within the catalytic domain when comparing spectra from the full-length and truncated protein. Published amide

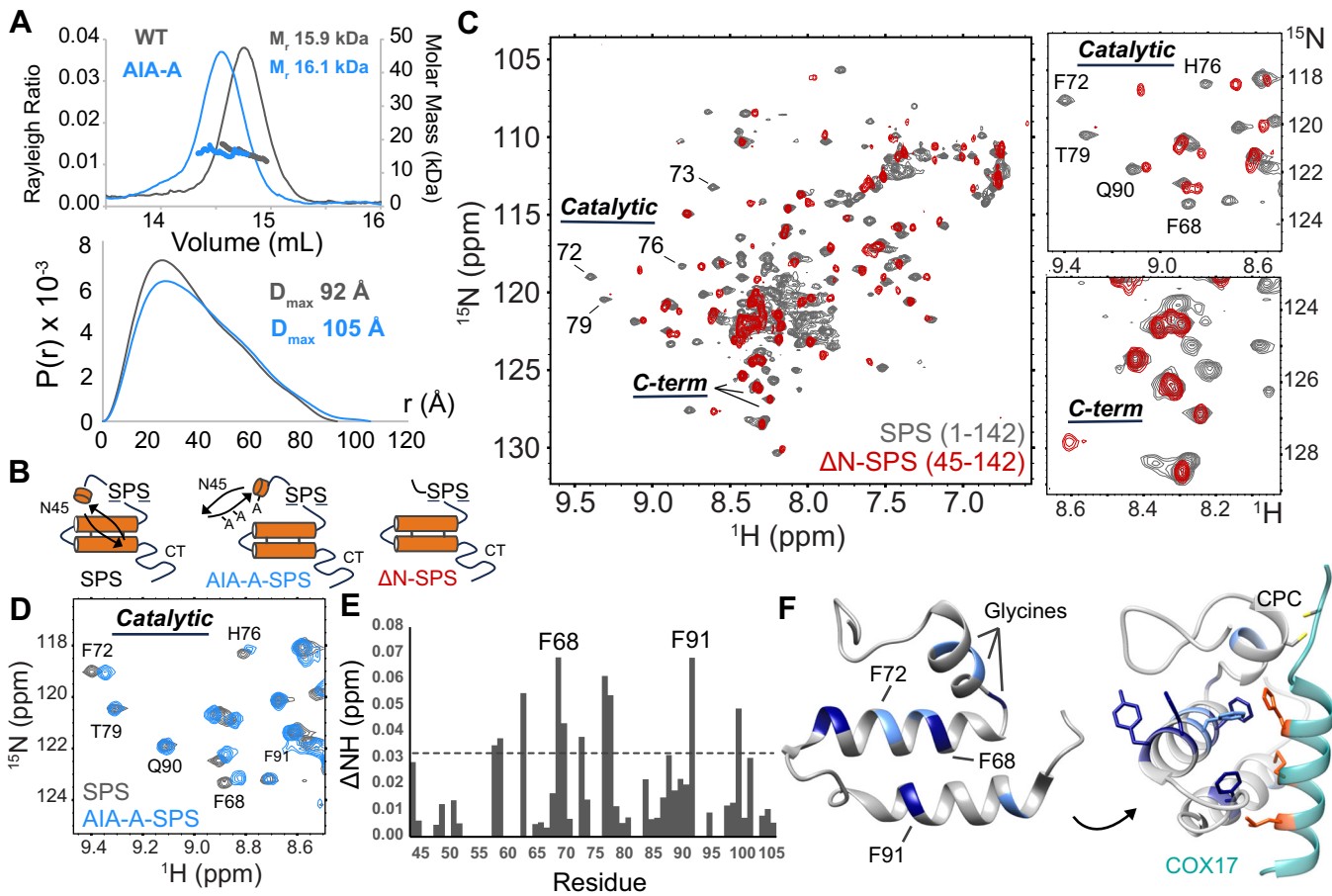

**Figure 3.** The AIF-interaction motif shields CHCHD4's redox-active site by mimicking unfolded CHCH substrates.

(A) The CHCHD4 AIA-A mutant exhibits an extended conformation by SEC-MALS (top) and SAXS P(r) distributions (bottom) relative to the wild-type protein. SEC-MALS is representative of two independent experiments. SAXS curves are representative of two to three independent experiments. (B) Diagrams of catalytically inactive CHCHD4 constructs used in NMR experiments. (C) Superimposed ¹H-¹⁵N-HSQC spectra from full-length (gray) and N-terminally truncated (red) inactive CHCHD4 (SPS). Expanded views highlight resonances from the active site (top) and C-terminal region (bottom). The NMR spectrum of full-length CHCHD4-SPS is representative of two independent experiments; the NMR spectrum of ΔN-CHCHD4-SPS represents one experiment. (D) Superimposed ¹H-¹⁵N-HSQC spectra from full-length CHCHD4-SPS (gray) and the inactive AIF-interaction defective mutant AIA-A-SPS (blue). NMR data is representative of two independent experiments. (E) Quantified HN chemical shift perturbations from the central CHCH domain of AIF-interaction defective mutant AIA-A relative to wild-type protein. The dotted line indicates the 2σ threshold. (F) Chemical shift perturbations exceeding 2σ (light blue) and 3σ (dark blue) are mapped onto the NMR structure of the CHCHD4 central domain (PDB: 2K3J). In the right panel, the central domain is displayed with the C-terminal helix of the COX17 substrate (teal) from the Mia40-COX17 complex (PDB: 2LOY). COX17 residues contacting the active site are highlighted in orange. Source data are available online for this figure.

backbone assignments from wild-type CHCHD4 (Banci et al, 2009) transfer well to the full-length CHCHD4-SPS ¹H-¹⁵N-HSQC spectrum, consistent with minimal structural disturbance from the inactivating SPS mutation (Appendix Fig. S5C). Superimposing ¹H-¹⁵N-HSQC spectra from CHCHD4 with and without the N-terminal AIF-interaction domain reveals widespread displacement of chemical shifts originating from the active site, supporting intramolecular contact between these domains (Fig. 3C). In contrast, backbone resonances originating from the flexible C-terminus remain unchanged, emphasizing its structural independence.

To assess if substrate mimicry is involved in the intramolecular shielding of CHCHD4's active site, we collected a ¹H-¹⁵N-HSQC spectrum on the inactive ¹⁵N-labeled β-hairpin/helix combination mutant (AIA-A SPS), which disrupts binding to AIF by targeting significant hydrophobic residues within the N-terminal domain.

Comparing this spectrum with that of the full-length protein reveals chemical shift perturbations for residues located within the catalytic domain (Fig. 3D; Appendix Fig. S5C). Quantifying and mapping significantly perturbed residues onto the CHCHD4 central domain highlights a cluster of aromatics on the active site surface (F68, F72, F91) that recognize and capture unfolded substrates (Banci et al, 2010; Habich et al, 2019; Sideris et al, 2009) (Fig. 3E,F). Perturbation mapping also highlights changes to the helical glycine hinge linking the catalytic CPC "lid" and the core substrate docking platform, suggesting its reorientation when the AIF-interaction motif is displaced from the active site.

These NMR results reveal an unexpected quaternary organization within human CHCHD4 and a substrate mimicry mechanism by which the AIF-interaction motif engages and shields the chaperone active site. Since CHCHD4's active site contains a hydrophobic binding surface and redox-sensitive CPC motif, this

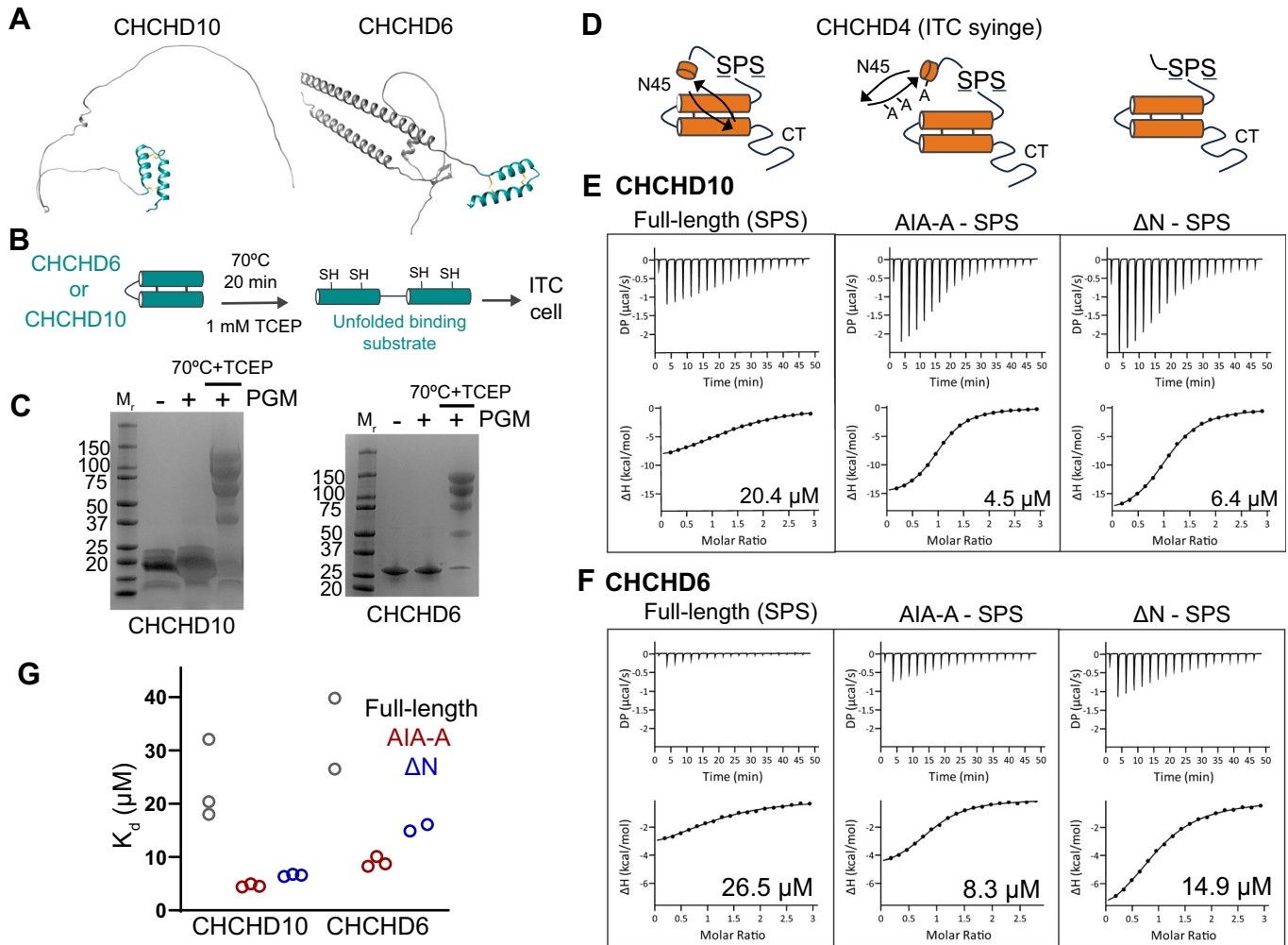

**Figure 4. The substrate-mimicking AIF-interaction motif regulates binding of CHCHD4 substrates.**

(A) CHCHD10 and CHCHD6 AlphaFold models highlighting their CHCHD4-targeted CX$_9$C helical hairpins (teal) See also Fig. EV4A. (B) Workflow for ITC reagent preparation. (C) SDS-PAGE of heat- and TCEP-treated CHCHD10 and CHCHD6 substrates labeled with PEG-MEM (PGM) confirms unfolding of their disulfide-stabilized helical hairpins. (D) Diagrams of catalytically inactive CHCHD4 constructs used in ITC experiments. (E) ITC thermograms, binding curves, and dissociation constants from titration of inactivated full-length (SPS), AIF-interaction defective (AIA-A-SPS), or N-terminally truncated (ΔN-SPS) CHCHD4 into unfolded CHCHD10 substrate. Results are representative of three independent experiments. (F) ITC thermograms, binding curves, and dissociation constants from titration of inactivated full-length (SPS), AIF-interaction defective (AIA-A-SPS), or N-terminally truncated (ΔN-SPS) CHCHD4 into unfolded CHCHD6 substrate. Results are representative of two to three independent experiments. (G) ITC-derived dissociation constants from inactive CHCHD4 mutant titrations into unfolded CHCHD10 and CHCHD6 substrates. See also Fig. EV4B. Source data are available online for this figure.

shielding mechanism may serve to protect the chaperone from adventitious interactions that could render it inactive. Moreover, as the AIF-interaction motif occludes the active site, this may serve as an underlying mechanism allowing AIF engagement to modulate CHCHD4 chaperone activity (Salscheider et al, 2022).

## The AIF-interaction motif regulates CHCHD4 substrate binding and chaperone activity

To determine if the AIF-interaction domain regulates CHCHD4 activity, we first investigated whether this domain influences CHCHD4 affinity for import substrates. Using isothermal titration calorimetry (ITC), we measured binding between inactive CHCHD4-SPS and unfolded CX$_9$C import substrates CHCHD10

and CHCHD6 (Figs. 4A and EV4A). Amyotrophic lateral sclerosis (ALS)-linked CHCHD10 is implicated in regulating mitochondrial dynamics and respiratory complex activity (Lehmer et al, 2018; Petel Legare et al, 2023; Ruan et al, 2022; Sayles et al, 2022; Straub et al, 2018), while CHCHD6 (MIC25) participates in the cristae-remodeling MICOS complex (Ding et al, 2015; Ott et al, 2015). These substrates were unfolded and reduced with heat treatment in the presence of 1 mM TCEP, and cysteine exposure was verified by PEG-conjugated maleimide (PEG-MEM) labeling (Fig. 4B,C). ITC thermograms demonstrate that both import substrates bind the full-length chaperone with comparatively weak affinity (K$_d$ 18–40 μM) (Fig. 4D–F). Strikingly, removing the N-terminus (ΔN) or neutralizing the AIF-interaction motif (AIA-A) increases affinity for both substrates (Fig. 4G). The low micromolar binding

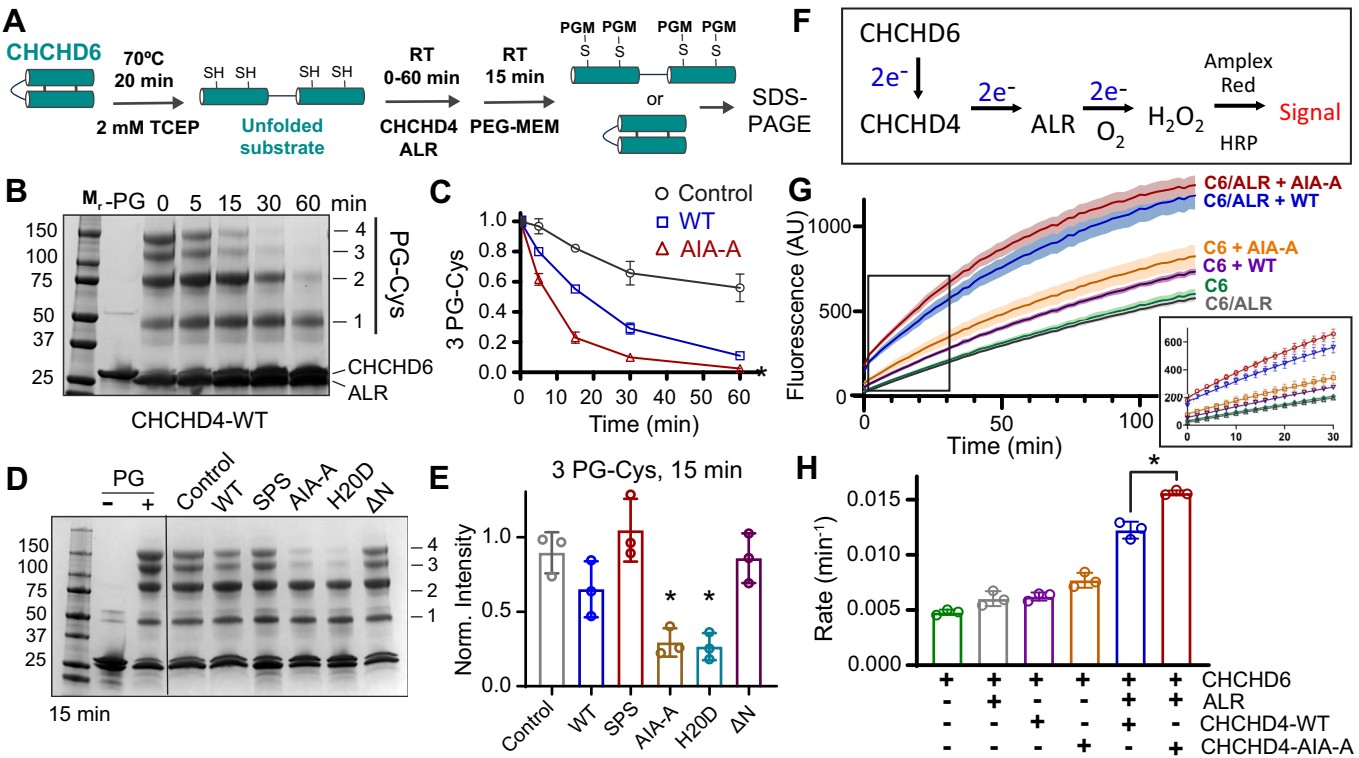

**Figure 5. The substrate-mimicking AIF-interaction motif regulates CHCHD4 chaperone activity.**

(A) Schematic overview of the CHCHD6 refolding assay. (B) CHCHD4 stimulates disulfide bond formation in vitro within unfolded CHCHD6 as monitored by PEG-MEM labeling and SDS-PAGE. See also Appendix Fig. S6A,B. (C) Quantified intensities from the 3-PG-cysteine band of chaperone assays in (B). Intensities are normalized to 0 min. Error bars reflect standard deviations and are centered at the average of 3 independent experiments. The 60-min control timepoint contains $n = 2$. (D) In vitro CHCHD4 chaperone assays monitored at 15 min for the panel of CHCHD4 mutants. (E) Quantified intensities from the 3-PG-cysteine band of chaperone assays in panel (D). Intensities are normalized to 0 min. Error bars reflect standard deviations and are centered at the average of three independent experiments. Asterisks indicate intensities that differ significantly relative to wild-type CHCHD4 with $P < 0.05$ by unpaired, two-tailed $t$ test with Welch's correction: CHCHD4-AIA-A ($P = 0.033$) and CHCHD4-H20D ($P = 0.016$). (F) Reaction scheme for the Amplex Red hydrogen peroxide fluorescent assay. Solvent molecular oxygen reacts with reduced ALR to generate hydrogen peroxide, which in turn reacts with Amplex red dye via horseradish peroxidase (HRP) to produce a fluorescent signal. (G) Time-resolved Amplex red fluorescent monitoring of hydrogen peroxide formation in CHCHD4 chaperone reactions. Each trace represents the average of three technical replicates with lighter error bands representing their standard deviation. The inset graph displays an expanded view of the early phase of the reaction with fitted curves. Results are representative of three independent experiments. C6, CHCHD6; WT, CHCHD4-WT; AIA-A, CHCHD4-AIA-A. (H) Quantification of signal rates from (G). Traces were fit with a single increasing exponential function. Error bars reflect standard deviations and are centered at the average of three independent experiments. The asterisk indicates significant difference between reactions ($P < 0.05$ by unpaired, two-tailed $t$ test with Welch's correction) containing wild-type CHCHD4 and the CHCHD4-AIA-A mutant ($P = 0.0116$). Source data are available online for this figure.

constants associated with these mutants are consistent with previously reported measurements between yeast Mia40 and CX₃C substrate Tim10 (Sideris et al, 2009). Thus, the AIF-interaction domain can regulate CHCHD4 affinity for import substrates.

We next probed whether the AIF-interaction domain can regulate CHCHD4 disulfide chaperone activity by using PEG-MEM to monitor in vitro disulfide formation within an unfolded CHCHD6 substrate (Fig. 5A). While CHCHD6 can spontaneously form internal disulfides within hours at room temperature, addition of active, wild-type CHCHD4 and regenerating enzyme ALR introduces a chaperoned disulfide for all CHCHD6 molecules within 60 min (Fig. 5B,C). As a control, the catalytically inactive SPS mutant fails to stimulate CHCHD6 disulfide formation over the spontaneous background rate (Fig. 5D). Mutating the AIF-interaction motif (AIA-A, H20D) significantly increases chaperone activity relative to the wild-type protein (Fig. 5D,E; Appendix

Fig. S6A,B), correlating increased binding affinity with more efficient chaperone function.

In parallel, we also monitored the impact of CHCHD4's AIF-interaction motif on electron flux through the disulfide relay, using Amplex Red fluorescence to detect the formation of hydrogen peroxide as a final electron acceptor (Tienson et al, 2009) (Fig. 5F–H; Appendix Fig. S6C). Consistent with the gel-based refolding assays, CHCHD6 produces a basal signal that is enhanced in the presence of the CHCHD4 chaperone and regenerating ALR. Substitution of wild-type CHCHD4 with the AIF-interaction mutant CHCHD4-AIA-A significantly enhances the rate of peroxide production (Fig. 5H), consistent with its increased chaperone activity observed in the refolding assays.

Notably, removing the N-terminus does not enhance CHCHD4 chaperone activity. In fact, the truncation mutant exhibits the marginal activity of the inactive SPS mutant (Fig. 5D,E). Unlike ΔN-CHCHD4-SPS samples used in the NMR and ITC experiments,

this construct retains a redox-sensitive CPC motif. Since the refolding assay is performed in the absence of reducing agent, we considered whether ΔN-CHCHD4 molecules form or retain inactivating intermolecular disulfide bonds, which would prevent successful engagement with substrates. To test this, we compared the elution of ΔN-CHCHD4 previously exchanged into βME-free refolding buffer and ΔN-CHCHD4-SPS by analytical size-exclusion chromatography, while maintaining an absence of reducing agent (Appendix Fig. S7A). While ΔN-CHCHD4-SPS elutes as a single peak, redox-active ΔN-CHCHD4 elutes as two species. The later peak aligns with monomeric ΔN-CHCHD4-SPS, while the leading peak suggests a disulfide-mediated dimer. Consistent with this, when 5 mM β-mercaptoethanol (βME) is included in the SEC running buffer, the fraction of monomeric ΔN-CHCHD4 increases (Appendix Fig. S7B). However, the intermolecular disulfide species remains largely unresolved and persists even through pre-incubation with 50–100-fold molar excess TCEP prior to SEC elution with or without βME (Appendix Fig. S7C,D). Complementary investigation of ΔN-CHCHD4 by non-reducing SDS-PAGE also confirms a mixture of unreacted and disulfide-linked ΔN-CHCHD4 that is only partially resolved by TCEP treatment (Appendix Fig. S7E).

Together, these results highlight CPC sensitivity to intermolecular disulfide formation and the unexpected stability of the resulting disulfide-mediated dimers. This reinforces a functional rationale for intramolecular shielding of CHCHD4 and explains the diminished in vitro activity of the truncated chaperone. However, CHCHD4 intramolecular shielding limits intrinsic substrate binding and chaperone activity of the full-length wild-type protein. The enhanced activity of the CHCHD4 AIF-interaction mutants AIA-A and H20D points to AIF association as a mechanism for sequestering the N-terminus and providing regulated access to the CHCHD4 active site.

### Activated AIF enables CHCHD4 substrate binding and stimulates chaperone activity

To determine whether AIF regulates CHCHD4 substrate access, we tested the ability of NADH-activated AIF to stimulate binding between catalytically inactive CHCHD4-SPS and an unfolded CHCH substrate fluorescently labeled for MST, Atto488-ΔN-CHCHD4-SPS (Fig. 6A,B). We selected CHCHD4's own unfolded catalytic CHCH domain as a substrate due to its compact size, which facilitates detection in MST binding assays, and to demonstrate CHCHD4's role in mediating its own chaperoned import in *trans*. The construct's removal of the N-terminal AIF-interaction domain and mutagenic inactivation of the CPC motif isolates the core CHCH domain for monitoring.

When co-incubated with either the unstimulated AIF monomer or an NADH control, CHCHD4-SPS fails to bind the unfolded fluorescent substrate (Fig. 6C). However, when NADH-activated AIF is included in the reaction, CHCHD4-SPS robustly binds Atto488-ΔN-CHCHD4-SPS. The addition of the isolated AIF-interaction domain (CHCHD4-N45) to the binding reaction is expected to competitively bind activated AIF, thus preventing CHCHD4-SPS engagement with AIF and chaperone deshielding. When it is included in the reaction, binding between CHCHD4-SPS and unfolded Atto488-ΔN-CHCHD4-SPS is decreased, supporting a role for AIF in regulating access to the CHCHD4 active site.

To assess if AIF stimulation enhances CHCHD4 chaperone activity, we monitored refolding of CHCHD6 in the presence of NADH-activated AIF (Fig. 6D–F). While co-incubation with either AIF or NADH alone does not alter wild-type chaperone activity, addition of NADH-activated AIF significantly accelerates CHCHD6 disulfide formation. Consistent with the effects seen for CHCHD4 substrate binding, the inclusion of the isolated AIF-interaction domain (CHCHD4-N45) hinders AIF's stimulation of chaperone activity (Fig. 6E,F). Thus, CHCHD4 association with NADH-activated AIF stimulates both CHCHD4 substrate binding and chaperone activity, providing a direct regulatory link between the chaperone activity of the disulfide relay and cellular availability of NADH. These results additionally highlight how AIF sustains CHCHD4 proteostasis through enhanced import and folding of nascent CHCHD4 polypeptides.

## Discussion

Mitochondrial import pathways tailor the mitochondrial proteome to metabolic demand and environmental stressors, providing fundamental support to cellular homeostasis. Multiple regulatory mechanisms link mitochondrial protein transport to centralized cellular sensors, modulating delivery of nascent proteins (Gerbeth et al, 2013; Harbauer et al, 2014; Priesnitz and Becker, 2018). While cellular regulation of other mitochondrial import pathways is relatively well-studied, how NADH-activated AIF and its structural allostery functionally support CHCHD4 persistence and import activity has remained enigmatic. Here we find that CHCHD4's scaffolding of dimeric AIF unveils a dedicated, NAD(H)-regulated import platform within the IMS that anchors and orients CHCHD4 to encounter incoming substrates, while also unshielding the CHCHD4 active site to stimulate disulfide chaperone activity (Fig. 7). This platform is enabled by the conserved, substrate-mimicking N-terminal hydrophobic β-hairpin/helix motif of CHCHD4's AIF-interaction domain, which serves dual roles in protecting the redox-sensitive active site and engaging AIF.

Though the AIF-CHCHD4 chimeric fusion used in structure determination bears the caveats of an engineered system, biochemical and cellular testing robustly support the revealed binding interface and its functional relevance for CHCHD4 chaperone activity. During final preparation of this manuscript, a low-resolution structure of a truncated CHCHD4 peptide bound to a mouse AIF dimer was reported (Fagnani et al, 2024). This structure nicely captures the β-hairpin of the AIF-interaction domain and its pairing to AIF's C-terminal domain, reinforcing our findings on the more complete human complex. However, the shorter murine CHCHD4 peptide fails to fully scaffold AIF. Consequently, it does not reveal the contacts of the anchoring histidine helix (H20) or the extended scaffolding of the N-terminus. Our data find that these are both important for binding and subsequent positioning of the CHCH catalytic domain toward the outer mitochondrial membrane. Moreover, combining our more detailed and comprehensive human structural data with biochemical and cytological results enabled insights on functional interactions linking NAD(H) sensing with protecting, regulating, and activating the CHCHD4 active site—the first reported mechanism for NAD(H)-dependent modulation of CHCHD4.

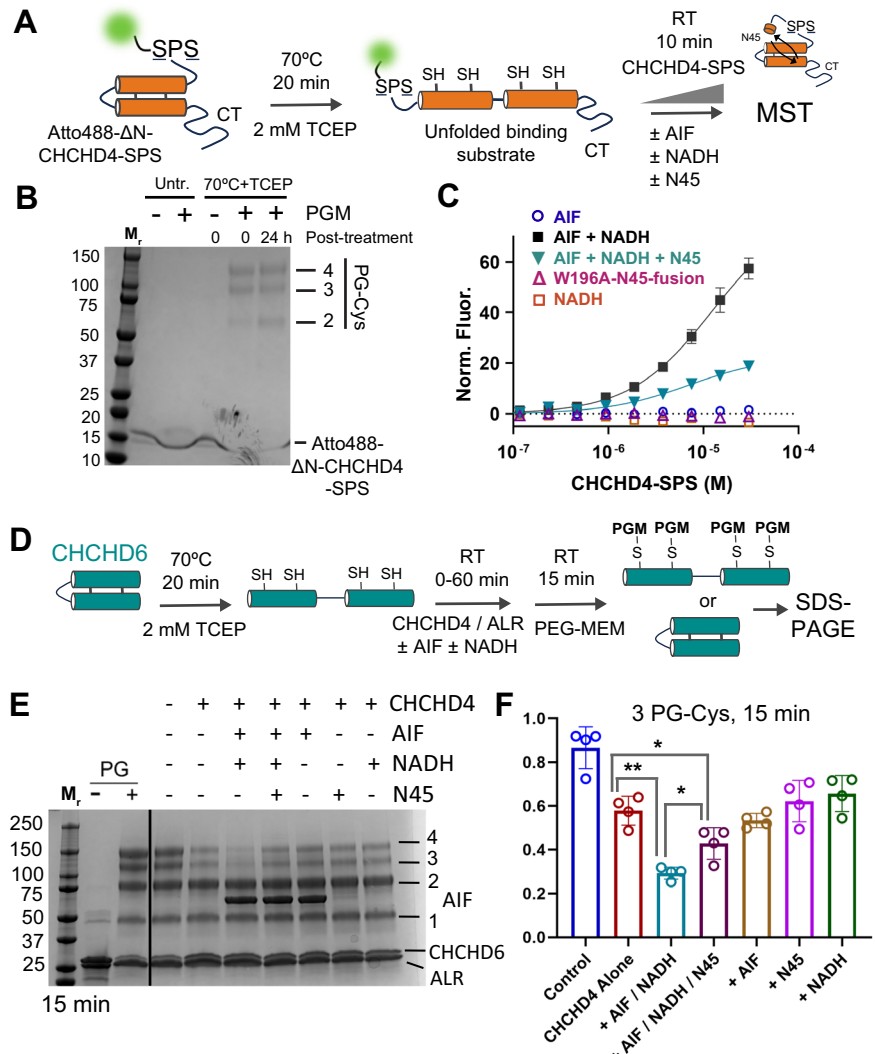

**Figure 6.  AIF engagement of CHCHD4's regulatory domain activates CHCHD4 substrate binding and chaperone activity.**

(A) Schematic of reagent preparation for MST binding experiments (B) SDS-PAGE of heat- and TCEP-treated Atto488-ΔN-CHCHD4-SPS substrate probed by PEG-MEM (PGM) confirms unfolding of its disulfide-stabilized CHCH hairpin. (C) Inactive CHCHD4-SPS binds the unfolded CHCH chaperone domain in *trans* when titrated in the presence of NADH-activated AIF. Each curve represents the average of three thermophoresis scans of a representative binding titration into Atto488-ΔN-CHCHD4-SPS. Binding titrations were repeated as least three times. (D) Schematic of the CHCHD6 refolding assay in the presence of AIF. (E) In vitro CHCHD4 chaperone assay monitoring CHCHD6 disulfide formation at 15 minutes in the presence of NADH-activated AIF. (F) Quantified intensities from the 3-PG-cysteine band of chaperone assays in (E). Intensities are normalized to 0 min. Error bars reflect standard deviations and are centered at the average of four independent experiments. Asterisks indicate intensities that differ significantly between indicated pairs with $P < 0.01$ (**) or $P < 0.05$ (*) by unpaired, two-tailed $t$ test with Welch's correction: CHCHD4 alone versus AIF-NADH ($P = 0.0013$), CHCHD4 alone versus AIF-NADH-N45 ($P = 0.0223$), and AIF-NADH versus AIF-NADH-N45 ($P = 0.0255$). Source data are available online for this figure.

The functional importance and conservation of CHCHD4's anchoring histidine helix suggests a potential role for pH in regulating AIF-CHCHD4 binding. With a side chain pKa ~6, histidine residues often provide a pH sensing function (Imenez Silva et al, 2022; Schonichen et al, 2013; Sisignano et al, 2021) driven by protonation and deprotonation of the imidazole ring. Since CHCHD4 H20 inserts into a hydrophobic pocket within AIF's C-terminal domain, protonation, which would create a positively charged side chain, is expected to cause a repulsive interaction. Cellular stressors, such as

hypoxia and metabolic dysregulation, can lead to decreased intracellular pH (Casey et al, 2010; Schonichen et al, 2013). Thus, pH-driven modulation of the AIF-CHCHD4 interaction and reduction of disulfide relay import activity could provide an effective autonomous dynamic means to temper OXPHOS expansion under stress conditions. Moreover, transient, local changes in IMS proton flux and gradients mediated by respiratory complex activity may provide a more nuanced layer of regulation to CHCHD4 import activity via this conserved interface histidine.

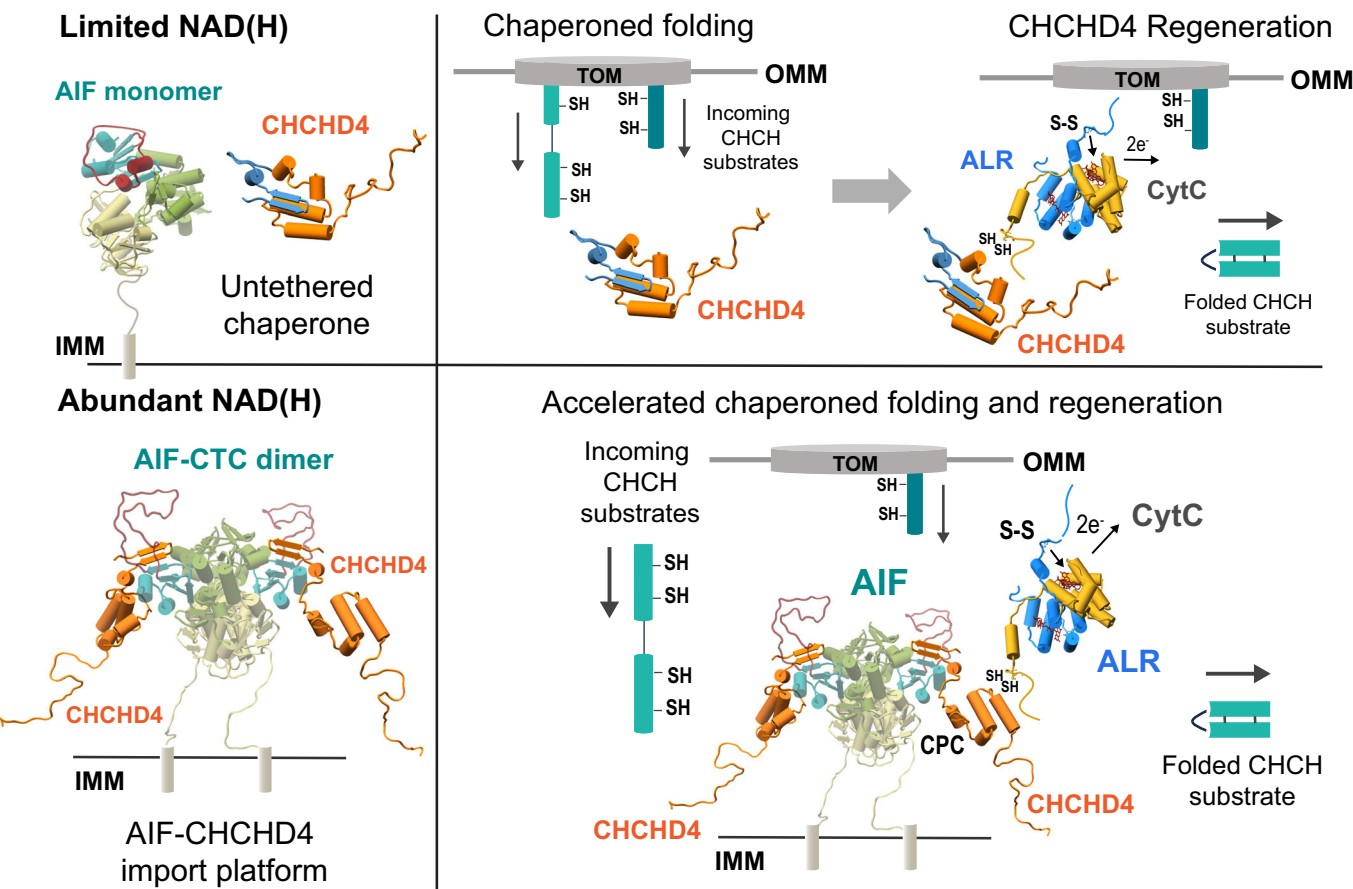

**Figure 7.  NADH-regulated mechanism of AIF stimulation of CHCHD4 chaperone activity.**

Under NAD(H) limited conditions, the CHCHD4 active site is shielded by the N-terminal AIF-interaction domain, resulting in diminished chaperone activity and import of CHCH substrates. When NADH is abundant, AIF is activated by NADH reduction, releasing its surface C-loop to create a binding surface for CHCHD4, thus unshielding the chaperone's active site. This enhances chaperone activity and import of CHCH substrates and may pair CHCHD4 chaperones to optimally interact with dimeric ALR for redox regeneration. Models are constructed from the reported crystal structure and PDB entries 4BV6 (AIF monomer), 3MBG (ALR), and 2K3J (CHCHD4 core).

The dimeric arrangement of the AIF-CHCHD4 complex notably complements the dimeric arrangement of regeneration partner ALR. Whereas ALR engages a single CHCHD4 molecule during regeneration, its proximity may be maintained by alternate hand-off between opposing CHCHD4 molecules on the AIF dimer platform, thereby increasing the efficiency of substrate import (Fig. 7). Notably, ALR's redox-active regeneration helix also uses a substrate-mimicry mechanism to engage the CHCHD4 active site and re-oxidize the catalytic CPC motif (Banci et al, 2011). Thus, the AIF dimer may serve as a broader organizational center for the CHCHD4 disulfide relay pathway by enabling orientation, activation, and regeneration of its components.

Importantly, CHCHD4's AIF-interaction domain protects the redox-sensitive catalytic CHCH domain through a novel intramolecular regulatory mechanism. CHCHD4 mutants lacking the AIF-interaction domain form intermolecular disulfide dimers resistant to chemical reduction, underscoring the need to shield the hydrophobic catalytic surface and redox-active CPC motif. By serving as a molecular mimic of hydrophobic CHCHD4 substrates,

the AIF-interaction domain engages with and forms an intramolecular shield over the central CHCH domain. Unfolded, nascent $CX_3C$ and $CX_9C$ substrates, such as CHCHD6 and CHCHD10 reported here, presumably compete with the AIF-interaction domain to access the active site and undergo chaperoned disulfide formation. When the domain is perturbed (AIA-A mutant), removed (ΔN mutant), or sequestered (interaction with NADH-activated AIF), substrate binding affinity and chaperone activity are markedly increased. Consequently, linking CHCHD4 unshielding to association with NADH-activated AIF ensures exposure of the chaperone active site only when CHCHD4 is securely anchored to AIF, thus regulating and limiting adventitious disulfide formation.

As AIF-CHCHD4 association stimulates CHCHD4 chaperone activity, the complex furthermore enables mitochondrial import of nascent CHCHD4 itself, since the catalytic CHCH domain requires chaperoned disulfide formation during its translation. Previous studies document decreased CHCHD4 abundance following AIF loss (Hangen et al, 2015), but it has been unclear whether this results from a novel import function provided by AIF or from

disturbance of the disulfide relay pathway. We tested CHCHD4's AIF-interaction domain as a novel mitochondrial localization signal by observing the cellular distribution of mutant CHCHD4 with a disrupted AIF-interaction motif (I12A/F14A/H20A), as well as a separate GFP-fusion of the AIF-interaction domain. Yet, disrupting the AIF-interaction motif did not hinder CHCHD4 targeting to mitochondria, nor did attaching CHCHD4's AIF-interaction domain to GFP result in mitochondrial localization. Thus, direct contact with AIF is not required for CHCHD4 to reach mitochondria, arguing against AIF as a novel import factor. In contrast, mutations disturbing CHCHD4's engagement with the disulfide relay through its own CX$_9$C recognition motif (SX9S) or hindering the disulfide relay pathway (SPS) resulted in CHCHD4 mitochondrial exclusion. Collectively, these results indicate that AIF's influence on CHCHD4 abundance arises through modulation of the disulfide relay import pathway. This notion is consistent with recent results noting variable responses in AIF-dependent CHCHD4 loss across different cells lines and CHCHD4 over-expression as a means to compensate for AIF loss (Habich et al, 2019; Salscheider et al, 2022).

Metazoan CHCHD4 regulation and AIF pairing distinctly contrasts with the unpaired organization of fungal Mia40. *S. cerevisiae* Mia40 has been a major model system for investigating mitochondrial import by the disulfide relay system. *S. cerevisiae* Mia40 and other fungal homologs lack the regulatory N-terminus observed in higher metazoans. Instead, the CHCH catalytic domain is directly anchored to the inner mitochondrial membrane via a single-pass transmembrane helix. This anchoring may be sufficient to restrict Mia40 encounters to nascent substrates and hinder adventitious CPC inactivation. As yet undiscovered internal Mia40 regulatory motifs and redox protective mechanisms unique to these single-cell organisms may also act to preserve the chaperone's catalytic motif. *S. cerevisiae* NADH-ubiquinone reductases Ndi1 and Nde1/Ndh1 and putative oxidoreductase Aif1 have been reported as fungal AIF homologs (Herrmann and Riemer, 2021). However, their architectures exhibit truncated C-terminal domains and lack AIF's regulatory C-loop and dimerization surface (Iwata et al, 2012; Jumper et al, 2021; Varadi et al, 2022), arguing against a comparable role in modulating Mia40.

That CHCHD4-AIF pairing is characteristic of multicellular organisms suggests their added capacity to tune the proteome of the disulfide relay to a wide variety of tissue-specific environments that rely on NADH metabolism. Since many CHCHD4 substrates support OXPHOS biogenesis and cristae formation (Modjtahedi et al, 2016), NADH-dependent AIF-CHCHD4 regulation of substrate flux appears suitable to tailor OXPHOS capacity to tissues with high or low metabolic demand, based on NADH available to support aerobic metabolism. This proposal also suggests a potential role for AIF-CHCHD4 regulation in mitochondrial biogenesis and ultimately tissue growth, as enhanced mitochondrial number and capacity are required for cell division. Our finding that optimal CHCHD4 activity requires NADH-activated AIF predicts that tissues and organs with high OXPHOS demands would rely more heavily on productive AIF-CHCHD4 association. Consistent with this prediction, symptoms in human patients with disease-causing AIF mutations are often localized to skeletal and cardiac muscle and the nervous system

(Ardissone et al, 2015; Berger et al, 2011; Ghezzi et al, 2010; Hu et al, 2017; Kettwig et al, 2015; Moss et al, 2021; Rinaldi et al, 2012).

The long-lived nature of the AIF-NADH charge-transfer complex (Churbanova and Sevrioukova, 2008) and the hydrophobic character of the AIF-CHCHD4 interface is consistent with CHCHD4's stable association with AIF in cells (Salscheider et al, 2022). Thus, one would anticipate a saturable response from the disulfide relay import pathway under conditions of abundant NADH and delayed reaction to declining NADH levels. Many disease-causing AIF mutants show reduced CTC lifetime or defective dimerization (Ghezzi et al, 2010; Rinaldi et al, 2012; Sevrioukova, 2016; Sorrentino et al, 2015), which may compromise stable, extended CHCHD4 interaction and stimulation. A destabilized AIF-CHCHD4 complex is thus expected to render the disulfide relay more sensitive to transient fluctuations in cellular NAD(H) and compromise sustained support for OXPHOS expansion during cell growth. This highlights a potential clinical role for small-molecule stabilizers of defective dimeric AIF, which could therapeutically reinforce AIF-CHCHD4 complexes (Brosey et al, 2024). Conversely, since inhibitors of respiratory Complex I can exert potent antitumor activity (Al Assi et al, 2024), hindering or reversing OXPHOS expansion through small-molecule destabilizers of native AIF-CHCHD4 complexes may provide opportunities for chemotherapeutics targeting tumor metabolic biochemistry and resistance (Das et al, 2024).

NAD(H) response dynamics may also be relevant to AIF's proposed role in the cell death pathway parthanatos (Fatokun et al, 2014). Here, AIF release from mitochondria follows rapid NAD$^+$ depletion arising from hyperactivation of the DNA damage signaling enzyme poly(ADP-ribose) polymerase 1 (PARP1) (Huang and Kraus, 2021). NAD(H)-coupled dissociation of AIF-CHCHD4 complexes during NAD$^+$ decline may influence the progression of AIF mitochondrial release. This also implies that therapies which modulate PARP1 signaling and NAD$^+$ consumption (Houl et al, 2019; Lord and Ashworth, 2017) may have secondary impacts on AIF-CHCHD4 mitochondrial function. Thus, knowledge of NAD(H) response dynamics related to the AIF-CHCHD4 complex could prove useful to understanding adverse mitochondrial effects associated with disease-causing AIF mutations and with stress- and age-related declines in NAD$^+$ levels.

Localization of the AIF-CHCHD4 import platform to sites of incoming mitochondrial substrates, such as the translocase of the outer membrane (TOM), is itself likely to be coordinated and regulated by additional accessory factors. Moreover, machinery associated with the trafficking and distribution of folded substrates may also interface with an AIF-CHCHD4 platform. Thus, continued investigation into novel interaction partners of AIF is likely to expand our understanding of the protein machinery regulating the mitochondrial proteome and the interplay between allosteric NAD(H) sensing by AIF, mitochondrial protein import, and cellular health. Delineating how the mitochondrial proteome is tuned to tissue-specific environments by AIF-NAD(H) metabolism and the disulfide relay is also expected to have implications for informed therapeutic interventions to modulate NAD(H) levels.

# Methods

### Reagents and tools table

| Reagent/resource | Reference or source | Identifier or catalog number |
|---|---|---|
| **Experimental models** | | |
| U2-OS cells (*H. sapiens*) | ATCC | HTB-96 |
| 293T cells (*H. sapiens*) | ATCC | CRL-3216 |
| **Recombinant DNA** | | |
| pANT7_cGST-CHCHD6 | DNASU Plasmid Repository | HsCD00640202 |
| pANT7_cGST-CHCHD10 | DNASU Plasmid Repository | HsCD00639923 |
| ALR codon-optimized G-block | Integrated DNA Technologies | |
| pET24b-AIF(78-613) | Brosey et al (2016) (https://doi.org/10.1016/j.str.2016.09.012) | |
| pET24b-AIF(104-613)-W196A | Brosey et al (2016) (https://doi.org/10.1016/j.str.2016.09.012) | |
| pET30a-CHCHD4 | Brosey et al (2024) (https://doi.org/10.1038/s41589-024-01609-1) | |
| Additional plasmids | This study | Dataset EV1 |
| **Antibodies** | | |
| Mouse anti-AIF, monoclonal | Santa Cruz Biotechnology | Cat# sc-13116 |
| Rabbit anti-HA, monoclonal | Cell Signaling Technology | Cat# 3724 |
| Rabbit anti-α-tubulin, monoclonal | Millipore Sigma | Cat# MABT522, clone RM113 |
| Horse anti-mouse IgG HRP | Cell Signaling Technology | Cat# 7076 |
| Goat anti-rabbit IgG HRP | Cell Signaling Technology | Cat# 7074 |
| Goat anti-rabbit IgG ATTO 550 | Millipore Sigma | Cat# 43328 |
| **Oligonucleotides and other sequence-based reagents** | | |
| PCR primers | This study | Dataset EV1 |
| **Chemicals, enzymes, and other reagents** | | |
| Q5® High-Fidelity DNA Polymerase | New England Biolabs | Cat# M0491S |
| Phusion DNA Polymerase | New England Biolabs | Cat# M0530S |
| Taq DNA Ligase | New England Biolabs | Cat# M0208L |
| T5 Exonuclease | New England Biolabs | Cat# M0363S |
| β-Nicotinamide adenine dinucleotide (NAD+) | New England Biolabs | Cat# B9007S |
| Deoxynucleotide (dNTP) Solution Mix | New England Biolabs | Cat# N0447S |
| Proteinase K | Worthington Biochemical Corporation | Cat# PROKRS |
| BS3 (bis(sulfosuccinimidyl) suberate) | Thermo Fisher Scientific | Cat# A39266 |
| β-Nicotinamide adenine dinucleotide, reduced disodium salt hydrate | Millipore Sigma | Cat# N8129 |

| Reagent/resource | Reference or source | Identifier or catalog number |
|---|---|---|
| mPEG-Maleimide, MW 10kDa | Laysan Bio | Cat# M-MAL-10K |
| Atto 488 NHS ester | Millipore Sigma | Cat# 41698 |
| Ammonium chloride ($^{15}$N, 99%) | Cambridge Isotope Laboratories, Inc. | Cat# NLM-467-5 |
| jetPRIME | Polyplus | Cat# 101000015 |
| MitoTracker™ Deep Red FM | Molecular Probes / Thermo Fisher Scientific | Cat# M22426 |
| Pierce™ Protease Inhibitor Mini Tablets, EDTA-free | Thermo Fisher Scientific | Cat# A32955 |
| Bio-Rad Protein Assay Dye Reagent Concentrate | Bio-Rad | Cat# 5000006 |
| Dynabeads™ Protein G for Immunoprecipitation | Thermo Fisher Scientific | Cat# 10003D |
| Clarity and Clarity Max ECL Western Blotting Substrates | Bio-Rad | Cat# 1705060S/1705062S |
| Bovine Serum Albumin | Sigma-Aldrich | Cat# A-4503 |
| Paraformaldehyde 16% Aqueous Solution EM Grade | Electron Microscopy Sciences | Cat# 15710 |
| Dulbecco's Modified Eagle's Medium-high glucose | Sigma-Aldrich | Cat# D6429-500ML |
| Gibco™ Fetal Bovine Serum, certified, heat-inactivated | Gibco / Thermo Fisher Scientific | Cat# 16140-071 |
| Gibco™ Antibiotic-Antimycotic (100X) | Gibco / Thermo Fisher Scientific | Cat# 15240-062 |
| Dulbecco's Phosphate Buffered Saline (1X) | Corning | Cat# 21-031-CV |
| **Software** | | |
| ASTRA 7.0.0 | Wyatt Technologies | |
| Clustal Omega | https://www.ebi.ac.uk/jdispatcher/msa/clustalo | |
| Espirit 3 | https://espript.ibcp.fr/ESPript/ESPript/ | |
| PR. Stability Analysis 1.0.2 (MST) | NanoTemper | |
| GraphPad Prism 9.0 | https://www.graphpad.com/ | |
| SAXS Frame Slice | https://sibyls.als.lbl.gov/ran | |
| ScÅtter 3.0 | https://bl1231.als.lbl.gov/scatter/ | |
| ATSAS Primus 3.0.1 | https://www.embl-hamburg.de/biosaxs/software.html | |
| SAXS Similarity | https://bl1231.als.lbl.gov/saxssimilarity/ | |
| BilboMD | https://bilbomd.bl1231.als.lbl.gov/ | |
| XDS (March 2019) | https://xds.mr.mpg.de/ | |
| CCP4 v 7.0 | https://www.ccp4.ac.uk/ | |

| Reagent/resource | Reference or source | Identifier or catalog number |
|---|---|---|
| Phenix 1.18.2 | https://phenix-online.org/ | |
| Phaser | https://www.mrc-lmb.cam.ac.uk/public/xtal/doc/phenix/reference/phaser.html | |
| Coot 0.8.9 | https://www2.mrc-lmb.cam.ac.uk/personal/pemsley/coot/ | |
| USCF Chimera 1.14 | https://www.cgl.ucsf.edu/chimera/ | |
| PyMOL 2.0 | Schrödinger | |
| Modeller 10.4 | https://salilab.org/modeller/ | |
| CellReporterXpress v 2.9.3 | Molecular Devices | |
| ImageJ 1.53K | https://imagej.net/ij/ | |
| Adobe Photoshop 2022 | Adobe | |
| Topspin 4.14 | Bruker | |
| Poky (Build 20220114) | https://sites.google.com/view/pokynmr | |
| MicroCal PEAQ-ITC Analysis Software (version 1.30) | Malvern Panalytical | |
| SoftMax Pro 7.1.2 | Molecular Devices | |
| GraphPad Prism 10.3.1 | Dotmatics/GraphPad Software, Inc. | |
| Bio-Rad Image Lab v 6.1.0 | Bio-Rad | |
| Other | | |
| Amplex Red Hydrogen Peroxide/Peroxidase Assay Kit | Invitrogen/Thermo Fisher Scientific | Cat# A22188 |

## Materials and reagents

G-blocks and primers used in subcloning were purchased from Integrated DNA Technologies (IDT). CHCHD6 (Uniprot Q9BRQ6) and CHCHD10 (Uniprot E5RGN4) constructs were obtained from the DNASU Plasmid Repository (Seiler et al, 2014; Yang et al, 2011). Enzymes for subcloning were obtained from New England Biolabs. ATTO-488-NHS ester for labeling MST reagents was purchased from Sigma-Aldrich (41698). Proteinase K was purchased from Worthington Biochemical Corporation. BS³ (bis(sulfosuccinimidyl)suberate) amine cross-linker was purchased from Thermo Scientific. NADH was purchased from Sigma-Aldrich. Methoxypolyethylene glycol maleimide (PEG-MEM) (10 kDa) was purchased from Laysan Bio and prepared according to the manufacturer's directions. $^{15}$N-labeled ammonium chloride was obtained from Cambridge Isotope Laboratories. Antibodies were purchased from Santa Cruz Biotechnology [AIF (sc-13116)], Cell Signaling Technology [HA (3724), HRP-conjugated mouse (7076) and rabbit (7074) secondary antibodies], Sigma-Aldrich Millipore [ATTO 550 rabbit polyclonal IgG (43328) and α-tubulin (MABT522, clone RM113)]. Mitotracker Deep Red FM dye was purchased from ThermoFisher/Life Technologies. Paraformaldehyde solution (16%) was obtained from Electron Microscopy Sciences.

## Cell culture

U2-OS (ATCC HTB-96) and 293T (ATCC CRL-3216) cells were obtained from ATCC and cultured in Dulbecco's modified Eagle's high-glucose (4.5 g/L) media (DMEM) (Sigma-Aldrich) with 10% (vol/vol) heat-inactivated fetal bovine serum (FBS) (Gibco) and 1% (vol/vol) antibiotic/antimycotic solution (Gibco) in a humidified incubator at 37 °C with 5% $CO_2$. Cells lines were tested for mycoplasma contamination upon receipt from ATCC.

## Plasmid construction

### Bacterial expression constructs

Wild-type AIF(78-613) and allostery mutants lacking the inner mitochondrial membrane tether have been reported previously (Brosey et al, 2016). CHCHD4 mutants were prepared from full-length CHCHD4 previously subcloned into a custom pET30a vector with an N-terminal GB1-6X-His fusion tag (Brosey et al, 2024). N-terminal truncation mutant ΔN-CHCHD4 (residues 40–142) and the N-terminal domain N45 (CHCHD4-N45, residues 1–45) were amplified from the full-length template and subcloned with pre-existing AgeI/Xho1 restriction sites into the pET30a vector. CHCDH4 point mutants (C53S/C55S (SPS), F12A/I14A (AIA), H20D, L28D) were constructed by Gibson assembly using the CHCHD4/pET30a template. CHCHD4 combination point mutant F12A/I14A/H20A (AIA-A) was prepared by introducing the H20A point mutant into the CHCHD4-AIA/pET30a template with Gibson assembly. Combination mutants ΔN-CHCHD4-SPS and CHCHD4-AIA-A-SPS were prepared by introducing C53S/C55S point mutations into ΔN-CHCHD4/pET30a and CHCHD4-AIA-A/pET30a templates with Gibson assembly.

To generate the chimeric fusion of AIF(104-613) W196A and CHCHD4(1-45), the CHCHD4 N-terminal domain was amplified with an N-terminal linker extension (SGSGPGSGS) and inserted at the C-terminus of AIF(104-613)-W196A contained in pET24b (Brosey et al, 2016) using Gibson assembly. CHCHD6 (Uniprot Q9BRQ6) and CHCHD10 (Uniprot E5RGN4) were obtained from the DNASU Plasmid Repository (Seiler et al, 2014; Yang et al, 2011) and inserted into pET30a by Gibson assembly. Full-length ALR (1–205) (Uniprot P55789) was purchased as a codon-optimized G-block from IDT, then amplified and inserted into pET30a with Age1/Xho1 restriction enzyme digestion. Neutral C154S/C165S mutations were introduced into ALR by site-directed mutagenesis to reduce disulfide-induced aggregation, as reported by Banci and colleagues (Banci et al, 2011). G-block and primer sequences used in subcloning are listed in Dataset EV1.

### Mammalian expression constructs

ALR was amplified from the G-block with a C-terminal GSS linker, HA epitope tag, and stop codon, then inserted into a pEGFP-N1 vector by Xho1/Age1 restriction enzyme digestion. Wild-type and mutant CHCHD4 were then amplified from bacterial expression constructs and inserted into this HA-pEGFP-N1 template by Gibson assembly. CHCHD4-SX9S (C64S/C74S/C87S/C97S) was created by combining a G-block insert carrying the serine mutations into the wild-type CHCHD4-HA/pEGFP-N1 background using Gibson assembly. The CHCHD4(1–45)-GFP fusion construct was generated by amplifying CHCHD4 (1–45) from its bacterial expression construct and subcloning into a

GFP-containing pEGFP-N1 plasmid at EcoR1/Age1 restriction digest sites.

## Protein expression and purification

AIF constructs and the AIF-CHCHD4 chimera were expressed in Rosetta2(DE3) cells at 37 °C in shake flasks with induction at $OD_{600}$ ~ 0.6 at 1 mM IPTG for 3 h (Brosey et al, 2016). CHCHD4 constructs, ALR, CHCHD6, and CHCHD10 were expressed following a similar protocol in Rosetta2Gami(DE3) cells to maintain an oxidizing environment favorable for disulfide formation. $^{15}N$-enriched CHCHD4 constructs were expressed in Rosetta2(DE3) in minimal media supplemented with 0.5 g $^{15}NH_4Cl/L$ in shake flasks as described above. Harvested bacterial pellets were stored at −20 °C until purification.

For protein purification, thawed bacterial pellets were resuspended in lysis buffer (25 mM HEPES, pH 8, 150 mM NaCl, 10-30 mM imidazole, 1 mM PMSF, 0.1% Tween-20, supplemented with 5 mM βME for all disulfide-containing constructs) and homogenized using a glass pestle. Lysates were sonicated (Qsonica), then clarified in a Beckman-Coulter JA25.50 rotor at 18 krpm (26,581 × g) for 20–30 min. Filtered lysates were passed over Ni-NTA resin (5 mL) (GE Healthcare/Cytiva) using gravity flow or loaded onto a pre-packed 5-mL HisTrap column with an ÄKTA Pure FPLC system. Columns were washed with 6–10 column volumes (CV) of Ni-NTA Buffer A (25 mM HEPES, pH 8, 150 mM NaCl, 30 mM imidazole ± 5 mM βME), then eluted isocratically (gravity flow) with 3–5 CV Ni-NTA Buffer B (Ni-NTA Buffer A with 300 mM imidazole) or by 10 CV gradient (FPLC). Protein-containing fractions were pooled and exchanged into dialysis buffer (25 mM HEPES, pH 7.5, 200 mM NaCl, 5 mM βME) overnight at 4 °C with PreScission protease to remove fusion tags. Dialysates were loaded onto subtractive Ni-NTA columns to separate cleaved tag from protein. Separated proteins were concentrated to 1–5 mL and loaded onto a HiLoad Superdex 200 16/600 column (AIF construct and AIF-CHCHD4 chimera) or a HiLoad Superdex 75 16/600 column (CHCHD4 constructs, ALR, CHCHD6, CHCHD10) equilibrated in 25 mM HEPES, pH 7.5, 150 mM NaCl, 5 mM βME. Protein-containing fractions were scanned by UV–Vis spectrophotometry to measure protein concentration and verify incorporation of AIF's FAD cofactor (for AIF constructs), then pooled, concentrated, and aliquoted for storage at −80 °C. See also Appendix Fig. S8 for representative examples of protein stocks.

## CHCHD4 multiple sequence alignment and phylogenetic tree

The CHCHD4 multiple sequence alignment and phylogenetic tree were generated with Clustal Omega (Madeira et al, 2022) using primary isoform sequences derived from the following Uni-Prot entries (UniProt, 2023): Q8N4Q1 (*H. sapiens*), Q8VEA4 (*M. musculus*), Q6DEI8 (*D. rerio*), Q9VAA6 (*D. melanogaster*), V6CJI3, (*C. elegans*), P36046 (*S. cerevisiae*), P87059 (*S. pombe*), O94030 (*C. albicans*), Q7S3S2 (*N. crassa)*, and Q4P8D2 (*U. maydis*). The CHCHD4 sequence alignment was formatted for display with Esprit 3 (Robert and Gouet, 2014).

## Microscale thermophoresis (MST) binding measurements

### *ATTO 488 protein labeling of wild-type AIF and CHCHD4 and mutants*

For MST experiments, purified AIF or CHCHD4 constructs were N-terminally labeled with ATTO 488-NHS ester (Sigma 41698).

1. Prepare a 2 mM stock of ATTO-488-NHS powered dye in DMSO, aliquot at 10 µL, and store at −80 °C.
2. Prepare labeling reaction buffer by combining equal parts SEC and sodium carbonate buffers: 750 µL 25 mM HEPES, pH 7.5, 150 mM NaCl and 750 µL 200 mM sodium carbonate, pH 9.
3. Dilute protein stock to a final concentration of 2 mg/mL in 300 µL labeling reaction buffer. Dilute ATTO-488-NHS dye to a final concentration of 200 µM in 100 µL labeling reaction buffer. Combine the two, cover with foil, and incubate for 1 h at room temperature. The final dye concentration should be at a twofold to threefold molar excess relative to the protein.
4. While the labeling reaction is incubating, equilibrate a bench-top PD-10 (Sephadex G-25) gravity-flow column with 5 column volumes (1 CV = 5 mL) of storage buffer (25 mM HEPES, pH 7.5, 150 mM NaCl) or equilibrate a 5 mL FPLC HiTrap Desalting column (GE Healthcare/Cytiva) on an ÄKTA Pure protein purification system (GE Healthcare/Cytiva). Have Eppendorf tubes (PD-10) or a 96-well deep-well plate (HiTrap Desalting) prepared for fraction collection.
5. Collect the labeling reaction and load onto the bench-top PD-10 column or HiTrap desalting column and begin collecting fractions. As the reaction enters the column, labeled protein and unreacted label should separate. If using the PD-10 column, add 2.1 mL storage buffer once the sample has entered the column bed, then elute with 3.5 mL storage buffer, taking care to avoid collection of the later-eluting free dye. If using the FPLC HiTrap desalting column, scan fractions with a NanoDrop UV–Vis spectrophotometer to identify fractions for pooling.
6. Collect a UV–Vis absorbance scan on the final pool of labeled protein recording $A_{280}$ and $A_{500}$ readings to calculate protein concentration. To correct for dye absorbance contributions at 280 nm, a correction factor (CF) specific to ATTO 488 is applied:

$$c_{protein} = (A_{280} - CF_{280} * A_{500})/(\varepsilon_{280} * l)$$

where $c_{protein}$ is molar concentration of the labeled protein, $A_{280}$ and $A_{500}$ are measured absorbance at 280 and 500 nm, $CF_{280}$ (0.09) is the ATTO 488 correction factor for 280 nm, $\varepsilon_{280}$ is the protein extinction coefficient ($M^{-1} cm^{-1}$), and l is the path length (cm).

7. Calculate the final degree-of-labeling (DOL) (number of fluorophore molecules per labeled conjugate) as follows:

$$DOL = (A_{max} \times \varepsilon_{280})/((A_{280} - A_{500} \times CF) \times \varepsilon_{500})$$

where $\varepsilon_{500}$ is the ATTO 488 extinction coefficient at 500 nm (90,000 $M^{-1} cm^{-1}$).

8. Prepare 25-µL aliquots of ATTO 488-labeled protein, flash freeze with liquid nitrogen, and store at −80 °C.

### MST binding titrations with Atto488-CHCHD4 and wild-type NADH-activated AIF

1. Prepare MST reaction buffer: 25 mM HEPES, pH 7.5, 150 mM NaCl, 0.01% Tween-20 with 5 mM TCEP to prevent intermolecular disulfide formation between CHCHD4 molecules.
2. Prepare 200 μL of 200 nM Atto488-CHCHD4 (2×) with 600 μM NADH or buffer in MST reaction buffer.
3. Prepare 30 μL of 60 μM unlabeled wild-type AIF and generate a 15-point 1:1 dilution series by sequentially combining 15 μL protein stock with 15 μL MST reaction buffer.
4. Combine 10 μL 2× Atto488-CHCHD4 stock with 10 μL from each 2× AIF dilution stock or 10 μL buffer and incubate for 10 min at room temperature. The final concentrations of Atto488-CHCHD4 are 100 nM −/+ 300 μM NADH with 1.8 nM –30 μM AIF.

### MST binding titrations with Atto488-CHCHD4 and AIF allostery mutants

1. Prepare MST reaction buffer: 25 mM HEPES, pH 7.5, 150 mM NaCl, 0.01% Tween-20 with 5 mM TCEP to prevent intermolecular disulfide formation between CHCHD4 molecules.
2. Prepare 200 μL of 200 nM Atto488-CHCHD4 (2×) in MST reaction buffer.
3. Prepare 30 μL of 60 μM unlabeled wild-type or AIF mutants (2×) and generate a 15-point 1:1 dilution series by sequentially combining 15 μL protein stock with 15 μL MST reaction buffer.
4. Combine 10 μL 2× Atto488-CHCHD4 stock with 10 μL from each 2× AIF dilution stock or 10 μL buffer and incubate for 10 min at room temperature. Final concentrations of Atto488-CHCHD4 are 100 nM with 1.8 nM – 30 μM AIF allostery mutants.

### MST binding titrations with Atto488-AIF(78-613) and wild-type CHCHD4 and mutants

1. Prepare MST reaction buffer: 25 mM HEPES, pH 7.5, 150 mM NaCl, 0.01% Tween-20 with 5 mM TCEP to prevent intermolecular disulfide formation between CHCHD4 molecules.
2. Prepare 200 μL 2× AIF with 200 nM Atto488-AIF, 800 nM unlabeled AIF, and 80 μM NADH and pre-incubate 10 min at room temperature to form charge-transfer complexes.
3. Prepare 30 μL of 30 (or 60) μM unlabeled wild-type or CHCHD4 mutants (2×) and generate a 15-point 1:1 dilution series by sequentially combining 15 μL protein stock with 15 μL MST reaction buffer.
4. Combine 10 μL 2× Atto488-AIF stock with 10 μL from each 2× CHCHD4 dilution stock or 10 μL buffer and incubate for 10 min at room temperature. Final concentrations of Atto488-AIF(78-613) are 100 nM with 1.8 nM–15 (or 30) μM wild-type or mutant CHCHD4.

### MST measurements and analysis with NanoTemper Monolith NT.115

Binding reactions were loaded into standard silica capillaries (NanoTemper). Microscale thermophoresis (MST) measurements were acquired on a Monolith NT.115 system (NanoTemper, PR.ThermControl 2.1.6) at 25 °C with 20–30% LED power and

40% infrared excitation for 20 s with 3-s equilibration and 1-s recovery periods. Time-averaged amplitudes were calculated over a 1-s window during excitation, and three consecutive scans were averaged to generate final values. MST amplitudes were determined using NanoTemper PR.Stability Analysis 1.0.2 and exported to GraphPad Prism 9.0 for analysis.

## MST binding measurements for unfolded CHCH substrate

### Preparation of unfolded ATTO-488-ΔN-CHCHD4-SPS

1. N-terminally label ΔN-CHCHD4-SPS with ATTO 488-NHS as described above.
2. Prepare a 20-μL reaction of 27 μM Atto488-ΔN-CHCHD4-SPS in 25 mM HEPES, pH 7.5, 150 mM NaCl with 2 mM TCEP. Using a thermocycler, incubate at 70 °C for 20 min. Cool the reaction for 5–10 min at room temperature.
3. To assess exposure of reduced cysteines, combine 2 μL of the unfolding reaction with 8 μL unfolding buffer ( ~ 5 μM protein) and label with 670 μM PEG-maleimide for 15 min at room temperature. Quench the labeling reaction with 5 μL 5× SDS-PAGE loading buffer and visualize on Coomassie-stained SDS-PAGE gradient gels to confirm unfolding of the catalytic CHCH domain.
4. Unfolded ATTO 488-ΔN-CHCHD4-SPS stocks can be aliquoted and frozen at −80 °C or used immediately for MST binding experiments.

### MST binding titrations with ATTO-488-ΔN-CHCHD4-SPS and AIF-CHCHD4 complexes

1. Prepare MST reaction buffer: 25 mM HEPES, pH 7.5, 150 mM NaCl, 0.01% Tween-20 with 5 mM TCEP to prevent intermolecular disulfide formation between CHCHD4 molecules.
2. Prepare 200 μL of 200 nM ATTO-488-ΔN-CHCHD4-SPS (2X) stocks in MST reaction buffer with one of the following components and pre-incubate 10 min at room temperature.
   a. Buffer (control)
   b. 300 μM NADH
   c. 30 μM AIF (78-613)
   d. 30 μM AIF/300 μM NADH
   e. 30 μM AIF/300 μM NADH/30 μM CHCHD4-N45
   f. 30 μM AIF-CHCHD4 chimera/300 μM NADH
3. Prepare 30 μL of 60 μM unlabeled CHCHD4-SPS (2×) and generate an 11-point 1:1 dilution series by sequentially combining 15 μL protein stock with 15 μL MST reaction buffer.
4. Combine 10 μL 2× ATTO-488-ΔN-CHCHD4-SPS stock with 10 μL from each 2× CHCHD4-SPS stock or 10 μL buffer and incubate for 10–15 min at room temperature. Final concentrations of ATTO-488-ΔN-CHCHD4-SPS are 100 nM (with additions 2a–f at 1×) with 15 nM – 30 μM CHCHD4-SPS.

### MST measurements and analysis with NanoTemper Monolith NT.115

Binding reactions were loaded into standard silica capillaries (NanoTemper). Microscale thermophoresis (MST) measurements were acquired on a Monolith NT.115 system (NanoTemper,

PR.ThermControl 2.1.6) at 25 °C with 20–30% LED power and 40% infrared excitation for 20 s with 3-s equilibration and 1-s recovery periods. Time-averaged amplitudes were calculated over a 1-s window during excitation, and three consecutive scans were averaged to generate final values. MST amplitudes and bound fractions were determined using NanoTemper PR.Stability Analysis 1.0.2 and exported to GraphPad Prism 9.0 for nonlinear curve-fitting analysis.

## SEC-MALS analysis of AIF-CHCHD4 complex formation

Analytical size-exclusion chromatography coupled to multi-angle light scattering (SEC-MALS) was used to monitor AIF-CHCHD4 complex formation. Experiments were performed with a Superdex 200 10/300 column equilibrated in 25 mM HEPES, pH 7.5, 150 mM NaCl, 0.02% sodium azide at 0.4 mL/min connected to an Agilent Technologies 1260 Infinity II HPLC in-line with a DynaPro Nanostar DLS detector (Wyatt Technologies), a Dawn Heleos II MALS detector (Wyatt Technologies), and an Optilab T-rex refractive index detector (Wyatt Technologies). BSA calibration was performed with 100 µL 2 mg/mL protein (Sigma-Aldrich). Samples (100 µL) of AIF (33 µM), wild-type or mutant CHCHD4 (50 µM with 5 mM TCEP), or complexes (with 120 µM NADH) were incubated at room temperature for 10 min, then spun at 20 kg for 10 min to remove particulates before loading. ASTRA 7.0.0 analysis software (Wyatt Technologies) was used for data analysis and molecular mass calculations.

## Limited proteolysis and BS³ amine cross-linking assays

### AIF limited proteolysis with Proteinase K

1. Prepare reaction buffer: 25 mM HEPES, pH 7.5, 150 mM NaCl.
2. Prepare 20-µL reactions of 10 µM wild-type AIF, AIF-W196A, or AIF-W196A-CHCHD4(1-45) chimera with buffer or 100 µM NADH and incubate for 10 min at room temperature.
3. Prepare a 5 µg/mL stock of Proteinase K (Worthington Biochemical Corporation) in reaction buffer and add to a final concentration of 0.25 µg/mL per reaction (1 µL per reaction). Incubate an additional 20 min.
4. Combine 6 µL of each reaction with 10 µL 2× SDS loading buffer, boil for 1–2 min, and visualize by 4–15% gradient SDS-PAGE.

The reported results are representative of three independent experiments.

### AIF BS³ amine cross-linking assays

1. Prepare reaction buffer: 25 mM HEPES, pH 7.5, 150 mM NaCl.
2. Prepare 20-µL reactions of 10 µM wild-type AIF, AIF-W196A, or AIF-W196A-CHCHD4(1-45) chimera with buffer or 100 µM NADH and incubate for 10 min at room temperature.
3. Resuspend powdered 2-mg aliquots of bis(sulfosuccinimidyl) suberate (BS³) amine cross-linker (ThermoFisher) at 25 mM in reaction buffer. Dilute into each reaction to a final concentration of 1.25 mM (1 µL per reaction) and incubate for an additional 30 min.
4. Combine 6 µL of each reaction with 10 µL 2× SDS loading buffer, boil for 1–2 min, and visualize by 4–15% gradient SDS-PAGE.

The reported results are representative of three independent experiments.

## Protein crystallization and structure determination

Crystals of the AIF-W196A-CHCHD4(1-45) fusion (AIF-CHCHD4 chimera) were grown by hanging-drop vapor-diffusion in 0.1 M HEPES, pH 8, 0.1 M NaCl, 18% PEG3350 with seeding of 8–12 mg/mL protein solutions. Crystals were harvested, briefly soaked in cryoprotective solution (0.1 M HEPES, pH 8, 0.1 M NaCl, 20% PEG3350 + 15% ethylene glycol), then flash-cooled in liquid nitrogen. Synchrotron X-ray diffraction data were collected at the Advanced Light Source beamline 8.3.1 at Lawrence Berkeley National Laboratory (LBNL). X-ray data were processed with XDS at the beamline (v March 2019) (Kabsch, 2010) and CCP4 (v. 7.0) (Agirre et al, 2023), and structures were solved by molecular replacement with the Phaser module (Bunkoczi et al, 2013) in Phenix 1.18.2 using coordinates PDB: 5KVH (Brosey et al, 2016). Structures were refined with Phenix (Liebschner et al, 2019), and model building was done in Coot (v. 0.8.9) (Emsley et al, 2010). Favored and allowed Ramachandran populations were 96.8% and 3.2%, respectively. Structural coordinates and crystallographic structure factors have been deposited with the Protein Data Bank as 8VGY. Molecular visualization and analysis were done in PyMOL (The PyMOL Molecular Graphics System, Version 2.0, Schrödinger, LLC) or UCSF Chimera (v 1.14). Crystallographic and molecular visualization software was accessed through the SBGrid Consortium (Morin et al, 2013).

## Model of full-length AIF-CHCHD4 complex

The AIF-CHCHD4 model was built in Chimera using the crystal structure of the AIF-W196A-CHCHD4 chimera complex and first conformer from the NMR ensemble of the CHCHD4 catalytic domain (PDB: 2K3J) (Banci et al, 2009). The CHCHD4 C-terminus and linker connecting the catalytic and bound AIF-interaction domains were built with Chimera's Modeller extension (Fiser and Sali, 2003; Webb and Sali, 2016) interfacing with the Modeller web server (v.10.4). AIF C-loops, which are disordered in the crystal structure, were also inserted with Modeller.

## AlphaFold analysis of CHCHD4 homologs

AlphaFold models were accessed from the AlphaFold Protein Structure Database (Jumper et al, 2021; Varadi et al, 2022) as follows: CHCHD4 [*H. sapiens* (AF-Q8N4Q1-F1-model_v2), *M. musculus* (AF-Q8VEA4-F1-model_v2), *D. rerio* (AF-Q6DEI8-F1-model_v2), *D. melanogaster* (AF-Q9VAA6-F1-model_v2), *C. elegans* (AF-V6CJI3-F1-model_v2)], Mia40 [*S. cerevisiae* (AF-P36046-F1-model_v2), *S. pombe* (F-P87059-F1-model_v2), *C. albicans* (AF-O94030-F1-model_v2), *N. crassa* (AF-Q7S3S2-F1-model_v2), *U. maydis* (AF-Q4P8D2-F1-model_v2)], *H. sapiens* CHCHD6 (AF-Q9BRQ6-F1-model_v2), *H. sapiens* CHCHD10 (AF-Q8WYQ3-F1-model_v2), *S. cerevisiae* Aif1 (AF-P52923-F1-model_v4), *S. cerevisiae* Nde1/Ndh1 (NDH1-AF-P40215-F1-model_v4), *H. sapiens* ALR (AF-P55789-F1-model_v4). AlphaFold models were analyzed and displayed with Chimera (Pettersen et al, 2004).

## Small-angle X-ray scattering (SAXS) data collection, analysis, and modeling

All SAXS samples were prepared by analytical gel filtration to ensure aggregate-free samples. Proteins were loaded onto a Superdex 200 10/300 column (AIF(78–613), AIF-CHCHD4 chimera) or Superdex 75 10/300 column (CHCHD4-WT and mutants) at 0.4 mL/min with collection of 0.5 mL fractions. Peak protein fractions of Superdex 200 purifications were collected and diluted into concentration series of 3–6 mg/mL. Peak CHCHD4 fractions from Superdex 75 purifications were pooled (1.5–2.0 mL), concentrated to 3–7.3 mg/mL in 4-mL Amicon ultra centrifugal filter concentrators with 10-kDa molecular-weight cut-off limits (Millipore-Sigma), then diluted into concentration series of 0.6–7.3 mg/mL. Two independent series were prepared per construct. each with three concentrations. Samples were transferred into 96-well PCR plates with matched buffers eluted from the protein-free void volume or collected from concentration flowthroughs. Plates were flash-frozen in liquid nitrogen and shipped on dry ice for SAXS data collection at the SIBYLS beamline (12.3.1) (Classen et al, 2013; Hura et al, 2009) at the Advanced Light Source at LBNL.

Datasets for the AIF-CHCHD4 chimera were collected at 10 °C as 0.5 s exposures for 15 s (30 frames) at an X-ray wavelength of 1.27 Å and 2 m sample-to-detector distance corresponding to a scattering vector q range of 0.01–0.39 Å$^{-1}$ ($q = 4\pi*\sin(\theta)/\lambda$ where $2\theta$ is the scattering angle) with a Pilatus3 2M pixel array detector. Datasets for wild-type and mutant CHCHD4 were collected similarly but with 0.3 s exposures for 10 s (33 frames) at an X-ray wavelength of 1.23 Å. X-ray scattering images were reduced and integrated using automated, in-house scripts from SIBYLS. Two independent buffer-matched datasets were collected before and after each sample, then averaged and subtracted frame-by-frame from the corresponding sample dataset to yield a buffer-subtracted, time-resolved scattering dataset.

Scattering curves were inspected for signs of radiation damage over the time course and averaged using ScÅtter 3.0 (https://bl1231.als.lbl.gov/scatter/) or the SAXS Frame Slice web application (http://sibyls.als.lbl.gov/ran). The first 0.5-s frame from the AIF-CHCHD4 chimera was excluded from averaging due to poor background subtraction. Averaged scattering curves were inspected for signs of radiation damage and aggregation by Guinier analysis. I(0) and $R_g$ values and P(r) distributions were generated using ScÅtter 3.0 and the Primus module of ATSAS (3.0.1) (Franke et al, 2017). AIF-CHCHD4 chimera scattering data was compared to AIF monomer and dimer reference SAXS curves from AIF (121–613) in SIMPLE SCATTERING entry XSUTRZQL (Brosey et al, 2016; Murray et al, 2023) using volatility-of-ratio ($V_R$) in the SAXS Similarity web application ((Hura et al, 2013) (https://bl1231.als.lbl.gov/saxs-similarity/).

For BilboMD analysis, a model of the AIF-CHCHD4-N45 chimera was constructed from the crystal structure, using Chimera's Modeller extension (Fiser and Sali, 2003; Webb and Sali, 2016) interfaced with the Modeller web server (v.10.4) to add N-terminal, C-loop, AIF-CHCHD4-45 linker, and C-terminal residues. The model was parameterized in the CHARMM forcefield with the CHARMM-GUI (Jo et al, 2008; Kim et al, 2017) to ensure proper parameterization of the FAD cofactors. The resulting CHARMM parameter files were provided to the BilboMD web application (Pelikan et al, 2009) (https://bilbomd.bl1231.als.lbl.gov/) for model generation and minimal ensemble search (MES) against the experimental SAXS data.

## CHCHD4 immunoprecipitations and western blot analysis

Cultured 293T cells were seeded to 70–80% confluence in 12-well plates and transfected with 1 µg plasmid DNA with jetPRIME (Polyplus). Cells were cultured for 24 h, then rinsed and harvested into 1× PBS by shearing. Cells were spun 5 min at 500 × g at 4°C, then flash-frozen and stored at −20 °C. Cell pellets were thawed on ice, resuspended in 100–125 µL lysis buffer (25 mM HEPES, pH 7.5, 150 mM NaCl, 5 mM TCEP, 0.5 mM EDTA, 0.1% Tween-20, Pierce protease inhibitor cocktail [EDTA-free] (Thermo Fisher Scientific)) and incubated on ice for 15 min. Whole-cell lysates were centrifuged for 10 min at 10 kg to isolate soluble fractions. Protein concentration was determined by Bradford assay, and lysates were diluted to 1 mg/mL.

Immunoprecipitation reactions were prepared by combining 200 µL 1 mg/mL lysate with 0.5 µL HA rabbit monoclonal antibody (Cell Signaling Technology 3724) and incubated for 1 h at 4 °C. Reactions were subsequently combined with 10 µL of pre-rinsed Protein G Dynabeads (Thermo Fisher Scientific) and incubated at room temperature for 15 min. Beads were rinsed three times with 200 µL lysis buffer and transferred to fresh tubes, prior to resuspension and boiling in 15 µL 1× SDS loading buffer. IP reactions and 20 µg input samples (10%) were run on 4–20% SDS-PAGE, transferred to PVDF membrane, blocked in 5% milk for 45–60 min, and probed overnight with antibodies to AIF (mouse, 1:500, Santa Cruz Biotechnology sc-13116) and HA epitope tag (rabbit, 1:5000, Cell Signaling Technology 3724) followed by detection with HRP-conjugated mouse and rabbit secondary antibodies (1:10,000, Cell Signaling Technology 7076 and 7074) and Clarity or Clarity Max ECL substrate (Bio-Rad). Developed blots were imaged with a Bio-Rad Chemidoc imager. Blots were then re-probed with antibody to α-tubulin (rabbit, 1:5000, Sigma-Aldrich (Millipore) MABT522, clone RM113), re-developed and imaged. Results are representative of at least three independent experiments.

## Immunofluorescence analysis of CHCHD4 mutant cellular localization

U2-OS cells were seeded in 12-well plates at 70–80% density and transfected with HA-tagged CHCHD4 constructs as described above. Cells were harvested 24 h after transfection and reseeded in clear bottom, 96-well plates (Corning 3904) and cultured for an additional 24 h. Cells were labeled with 0.5 µM Mitotracker Deep Red FM (Molecular Probes) for 30 min (37 °C, 5% $CO_2$), rinsed with chilled 1× PBS, then fixed in 4% paraformaldehyde/1× PBS for 15 min at room temperature. Fixed cells were permeabilized with 0.5% Triton-X 100/1× PBS for 5 min on ice, then blocked with 2% BSA/1× PBS for 1 h at room temperature. Cells were probed overnight at 4 °C with monoclonal antibody to the HA epitope tag (rabbit, 1:1000, Cell Signaling Technology 3724), rinsed three times with 1× PBS, then probed with ATTO 550-conjugated rabbit polyclonal IgG secondary antibody (1:400, Sigma-Aldrich (Millipore) 43328) and 10 µg/mL Hoescht for 1 h at room temperature. Cells were rinsed three times with 1× PBS and imaged at 40× magnification on an ImageXPress Pico high-content cell imager using CellReporterXpress v 2.9.3.1183 imaging software (Molecular Devices). Images were exported as TIFF files, channels were merged with ImageJ 1.53 K (Schneider et al, 2012), and images were cropped in Adobe Photoshop 24.7.3. U2-OS cells expressing CHCHD4 (1–45)-GFP were transfected and processed as

described without antibody probes, then imaged at 20× magnification. Results are representative of at least three independent experiments.

## NMR spectroscopy

SEC fractions of $^{15}$N-enriched CPC-inactivated full-length (SPS), N-terminally truncated (ΔN-SPS), and AIF-defective (AIA-A-SPS) CHCHD4 were buffer exchanged into NMR buffer (25 mM HEPES, pH 7.0, 75 mM NaCl, 5 mM TCEP) with 4-mL Amicon centrifugal filters (10 kDa MWCO), then diluted to a final concentration of 30–200 μM with addition of 10% D$_2$O. Sensitivity-enhanced $^1$H-$^{15}$N-HSQC spectra were collected at the Rice University Keck NMR Facility on a Bruker NEO 600 MHz High-Performance Digital NMR with a helium-cooled inverse TCI 600S3 H&F/C/N-D-05 ZXT probe through the Shared Equipment Authority of Rice University (https://research.rice.edu/sea/). Data were processed in Topspin 4.14 and analyzed with Sparky modules provided in Poky (Build 20220114) (Lee et al, 2021). Published backbone resonance assignments for full-length CHCHD4 (Banci et al, 2009) were accessed from the Biological Magnetic Resonance Bank (BMRB: 17646) and transferred to the SPS mutant (Appendix Fig. S5C). Chemical shift perturbations (CSPs) were calculated using $\Delta HN = \sqrt{0.5 * [(H_A - H_B)^2 + 0.2 * (N_A - N_B)^2]}$ and analyzed following the method of Williamson (Williamson, 2013). Significant CSPs were identified by first calculating an initial standard deviation (σ) across all CSPs, then calculating a focused standard deviation (σ$_0$) with the exclusion of CSP values exceeding 3σ. CSPs exceeding 2σ$_0$ (ΔNH > 0.030) or 3σ$_0$ (ΔNH > 0.044) were considered significant for analysis. Two independent $^{15}$N-$^1$H-HSCQ NMR spectra were acquired for CHCHD4-SPS and AIA-A-SPS; due to sample limitations, a single spectrum was acquired for ΔN-SPS.

## Isothermal titration calorimetry

ITC affinity measurements were carried out with a MicroCal PEAQ-ITC Automated (Malvern Panalytical). Buffer-matched CPC-inactivated full-length (SPS), N-terminally truncated (ΔN-SPS), and AIF-defective (AIA-A-SPS) CHCHD4 stocks (700-750 μM in 25 mM HEPES, pH 7.5, 150 mM NaCl, 1 mM TCEP) were titrated into 50 μM unfolded CHCHD6 and CHCHD10 substrates at 25 °C in 18 2-μL injections (4 s each) spaced at 150 s and stirred at 750 rpm. Unfolded substrates were prepared by heating 50 μM protein with 1 mM TCEP at 70 °C for 20 min, then cooling to room temperature for 10–15 min. Substrate unfolding was assessed by PEG-MEM labeling of free cysteines (see below). Two (CHCHD6 + SPS, ΔN-SPS) or three independent ITC titrations were collected for each substrate/chaperone complex. ITC thermograms were baseline corrected and integrated in MicroCal PEAQ-ITC Analysis Software (version 1.30). The same software was used to determine binding constants and thermodynamic parameters.

## CHCHD4 in vitro chaperone assays

### Reagent preparation

To ensure homogeneous redox states among CHCHD4 chaperones, stocks of wild-type and mutant CHCHD4 are pre-treated with TCEP.

1. Retrieve purified wild-type or mutant CHCHD4 protein stocks (200–400 μM) and add 10 mM TCEP. Incubate for 20 min at room temperature to ensure complete reduction of the CPC motif.
2. Equilibrate a 5-mL HiTrap Desalting column (GE Healthcare/Cytiva) on an ÄKTA Pure protein purification system (GE Healthcare/Cytiva) with reaction buffer 25 mM HEPES, pH 7.5, 150 mM NaCl.
3. Once incubation is complete, load the reduction reaction onto the HiTrap Desalting column and collect fractions in a 96-deep well block.
4. Scan fractions for protein content with NanoDrop UV–Vis spectrophotometer, pool fractions with protein (usually 1.5–2.0 mL total), aliquot and flash freeze with liquid nitrogen for storage at −80 °C to maintain a reduced state.

To prepare unfolded CHCHD6 as a chaperone substrate, protein stocks are heat-treated in the presence of a reducing agent.

1. Retrieve purified CHCHD6 protein stocks (200–400 μM) and add 2 mM TCEP. Transfer to PCR tubes and incubate in a thermocycler for 20 min at 70 °C.
2. Load the heat-treated reaction onto a HiTrap Desalting column equilibrated in reaction buffer and collect fractions in a 96-deep well block.
3. Scan fractions for protein content with NanoDrop UV–Vis spectrophotometer, pool fractions with protein (usually 1.5–2.0 mL total), aliquot and flash freeze with liquid nitrogen for storage at −80 °C to maintain a reduced state.

ALR (C154S/C165S) should remain oxidized for the refolding reaction. Protein stocks are exchanged directly into reaction buffer.

1. Retrieve purified ALR (C154S/C165S) protein stocks (200–300 μM) and load onto HiTrap Desalting column equilibrated in reaction buffer, collecting fractions in a 96-deep well block.
2. Scan fractions for protein content with NanoDrop UV–Vis spectrophotometer, pool fractions with protein (usually 1.5–2.0 mL total), aliquot, and flash freeze with liquid nitrogen for storage at −80 °C.

### Refolding assays to assess CHCHD4 mutant activity

Refolding reactions are assembled with CHCHD4, ALR, and buffer, then refolding is initiated by the addition of unfolded CHCHD6.

1. Combine the following for each refolding reaction (15 μL):
   a. 2 μL 15 μM CHCHD4 (2 μM final)
   b. 5 μL 30 μM ALR (10 μM final)
   c. 3 μL reaction buffer
2. Initiate refolding by adding 5 μL 30 μM CHCHD6 (10 μM final) to each reaction and incubate at room temperature for the target time (0–60 min).
3. Add 1 μL 6.7 mM PEG-maleimide (stock prepared in DMSO) to each reaction (final concentration ~400 μM) and incubate at room temperature to label free cysteines.
4. Quench labeling with the addition of 5 μL 2× SDS loading buffer to each reaction, boil 1–2 min, and visualize on a 4–15% gradient SDS-PAGE with Coomassie staining.

### Refolding assays to assess AIF stimulation of CHCHD4 activity

Refolding reactions are assembled with CHCHD4, ALR, and buffer, then refolding is initiated by the addition of the specified AIF reagent and unfolded CHCHD6.

1. Combine the following for each refolding reaction (15 µL):
   - 2 µL 15 µM CHCHD4 (2 µM final)
   - 5 µL 30 µM ALR (10 µM final)
2. Prepare the following AIF and NADH stocks and incubate for 10 min to facilitate AIF-CTC formation, then add to each reaction:
   - 3 µL reaction buffer
   - 3 µL 80 µM NADH (16 µM final)
   - 3 µL 40 µM CHCHD4-N45 (8 µM final)
   - 3 µL 20 µM AIF (78-613) (4 µM final)
   - 3 µL 20 µM AIF/80 µM NADH
   - 3 µL 20 µM AIF/80 µM NADH / 40 µM CHCHD4-N45
3. Initiate refolding by adding 5 µL 30 µM CHCHD6 (10 µM final) to each reaction and incubate 15 min at room temperature for the target time (0–60 min).
4. Add 1 µL 6.7 mM PEG-maleimide (stock prepared in DMSO) to each reaction (final concentration ~400 µM) and incubate at room temperature to label-free cysteines.
5. Quench labeling with the addition of 5 µL 2× SDS loading buffer to each reaction, boil 1–2 min, and visualize on a 4–15% gradient SDS-PAGE with Coomassie staining.

### Quantification and statistical comparison of PEG-MEM-CHCHD6 reactions to evaluate refolding

SDS-PAGE gels were digitized with a Bio-Rad ChemiDoc imaging system. Intensities of protein bands representing CHCHD6 modified with 3 PEG-MEM molecules were quantified across each time series with Bio-Rad Image Lab software (version 6.1.0 build7) and normalized to intensity at 0 min. Normalized intensities were averaged across three (CHCHD4 mutants) or four (AIF-CHCHD4) independent experiments and reported with their standard deviations. Sample variances were assessed for similarity across reactions by F-test. As select sample variances demonstrated significant differences, testing for statistical significance between average intensities utilized an unpaired, two-tailed $t$ test with Welch's correction (provided by GraphPad Prism 9.0). Intensity differences were considered significant for $P$ values < 0.05; individual $P$ values are reported in the associated figure captions. Sample size estimation and blinding were not applicable to these experiments, as study design did not involve population-wide assessments with allocation to treatment groups.

## Amplex red hydrogen peroxide fluorescent assays

### Reagent preparation

To ensure homogeneous redox states among CHCHD4 chaperones, stocks of wild-type and mutant CHCHD4 and CHCHD6 are pre-treated with TCEP and exchanged into reaction buffer as described above. ALR (C154S/C165S) stocks are also pre-exchanged into reaction buffer. Reconstitute stocks of horseradish peroxidase (HRP) (10 U/mL in 50 mM sodium phosphate, pH 7.5, 1 mL) and Amplex™ Red dye (10 mM in DMSO, 60 uL per vial) according to the manufacturer's directions provided in the Amplex™ Red Hydrogen Peroxide/Peroxidase Assay Kit (Invitrogen / Thermo Fisher Scientific), Store aliquots of dye and enzyme stocks at −80 °C.

### FlexStation 3 MicroPlate reader protocol

Within SoftMax Pro 7.1.2 prepare a protocol for time-resolved monitoring of Amplex Red fluorescence in a 384-well plate with the following parameters:

- *Excitation: 530 nm / Emission: 590 nm*
- *Plate parameters: 384-well black, round well, flat bottom*
- *Read interval: 2 min*
- *Total read time: 2 h*

### Amplex Red fluorescent assay

The Amplex Red protocol was based on a modified version of that reported in (Tienson et al, 2009). Refolding reactions are initially assembled with CHCHD4 (wild-type or mutant), ALR, and reaction buffer in 96-well plates. Refolding is then initiated by adding the mixture into a 384-well plate containing unfolded CHCHD6 substrate, Amplex Red dye, and horseradish peroxidase (HRP), followed by immediate monitoring with the FlexStation 3 microplate reader. Premixing CHCHD6 with Amplex Red dye and HRP creates negligible background signal and minimizes dead time between reaction initiation and microplate readout. Final reactions contain 2 µM CHCHD4 (wild-type or mutant), 0.25 µM ALR, and 10 µM CHCHD6 with 0.1 U/mL HRP and 50 µM Amplex Red dye in a 40-µL reaction volume. Final ALR concentrations were selected to minimize non-specific background signal.

Prepare triplicate refolding reactions (40 µL ×3) as follows:

1. Pre-equilibrate CHCHD4 at room temperature for 30 min.
2. Prepare reactions by combining the following for each technical triplicate:
   - 6 µL 40 µM CHCHD4 (wild-type or mutant) or buffer
   - 6 µL 5 µM ALR or buffer
   - 24 µL reaction buffer
     *Each 36-µL triplicate reaction is prepared in a single well of a 96-well PCR plate and will eventually be distributed and mixed into a 384-well plate using a multi-channel pipette (12 µL per well).*
3. To prepare dye mixture, dilute 1/50 (v/v) HRP stock (final concentration 0.2 U/mL) and 1/100 (v/v) Amplex Red dye stock (final concentration 100 µM) in reaction buffer. For each triplicate sample combine the following to generate the dye/substrate mixture:
   - 60 µL HRP (0.2 U/mL): Amplex Red dye (100 µM) mixture
   - 24 µL 50 µM CHCHD6 (or buffer for control reactions)

   Dispense 28 µL of the dye/CHCHD6 mixture into each well of a 384-well plate, leaving every other column blank to facilitate transfer from the 96-well PCR plate.

4. Using a multi-channel pipette, quickly dispense and mix 12-µL CHCHD4/ALR mixture into wells containing the dye/CHCHD6 mixture to initiate the reaction. Insert the plate into the FlexStation 3 and begin reading immediately for a total of 2 h.
5. To analyze data, export the raw time-resolved fluorescent data into Excel format and calculate averages and standard deviations among technical replicates. Subtract averaged curves of buffer alone from each averaged trace.

6. Copy processed data into GraphPad Prism 10.3.1 and perform nonlinear regression curve-fitting using a one phase association exponential to generate rates for increasing fluorescence.

7. Averaged rates and standard deviations from three independent experiments are tested for statistical significance between WT and AIA-A CHCHD4 using an unpaired, two-tailed *t* test with Welch's correction for unequal variances. Intensity differences were considered significant for *P* values < 0.05; individual *P* values are reported in the associated figure captions. Sample size estimation and blinding were not applicable to these experiments, as the study design did not involve population-wide assessments with allocation to treatment groups.

### Analytical SEC analysis of intermolecular ΔN-CHCHD4 disulfides

Size-exclusion chromatography experiments were performed with a Superdex 75 10/300 column equilibrated in 25 mM HEPES, pH 7.5, 150 mM NaCl ± 5 mM βME on an ÄKTA Pure protein purification system. Initial 200-µL samples of 200 µM ΔN-CHCHD4 previously exchanged into βME-free refolding assay buffer and 200 µM ΔN-CHCHD4-SPS were eluted in the absence of βME at 0.4 mL/min. The ΔN-CHCHD4-SPS elution profile was consistent with previous SEC runs generated during at least two purifications. Succeeding 150-µL samples of 100 µM βME-free ΔN-CHCHD4 were treated with 0, 5, or 10 mM TCEP for 20 min at room temperature, then eluted in the absence of βME, or were treated with 0 and 10 mM TCEP and eluted in the presence of βME. Chromatograms were exported to Microsoft Excel (version 2303) for visualization.

### Non-reducing SDS-PAGE monitoring intermolecular ΔN-CHCHD4 disulfides

1. Prepare reaction buffer: 25 mM HEPES, pH 7.5, 150 mM NaCl.
2. Prepare 15-µL reactions of 20 µM wild-type ΔN-CHCHD4 combined with 0, 1, 2, 5, or 10 mM TCEP and incubate for 30 min at room temperature. Include two duplicates with 0 or 10 mM TCEP as controls.
3. Add 3 µL 5× SDS loading buffer *without* βME (or other reducing agent) to titration samples or 3 µL 5× SDS loading buffer with 40 mM βME to quench reactions and boil 1 min.
4. Run samples on a 4–15% SDS-PAGE gel and visualize with Coomassie staining.

Results are representative of at least two independent experiments.

## Data availability

The datasets produced in this study are available in the following databases: Crystallographic structure factors and coordinates: Protein Data Bank 8VGY (https://www.rcsb.org/). SAXS data: SIMPLE SCATTERING XS6ARGD0 (AIF-CHCHD4 chimera) and XS4FJZBK (wild-type and mutant CHCHD4) (https://simplescattering.com/) and the Small Angle Scattering Biological Data Bank (SASBDB) (https://www.sasbdb.org) SASDV47 (AIF-CHCHD4 chimera), SASDVY6 (CHCHD4 wild-type), SASDVZ6

(CHCHD4 F12A/I14A/H20A, 'AIA-A'), SASDV27 (CHCHD4 H20D), and SASDV37 (CHCHD4 L28D). Micrographs (Fig. 2D): The BioImage Archive (Sarkans et al, 2018) S-BIAD1515.

The source data of this paper are collected in the following database record: biostudies:S-SCDT-10_1038-S44318-024-00360-6.

## Peer review information

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

## Acknowledgements

The authors thank Dr. George Meigs of ALS beamline 8.3.1 for assistance with remote data collection and Dr. Quinn Kleerekoper for support at the Rice University Keck NMR facility. We thank Dr. Todd Link for assistance with protein purification, Dr. Zamal Ahmed for training related to data collection and analysis for MST and high-content imaging and Nicole Wright for assistance with MST experiments. This work was supported by National Institutes of Health (NIH) grants (P01 CA092584 (JAT), R35 CA220430 (JAT)), the Cancer Prevention and Research Institute of Texas (RP180813) (JAT), and a Robert A. Welch Chemistry Chair (G-0010) (JAT). All SAXS data was collected at the SIBYLS Advanced Light Source beamline (12.3.1) which operates through support from the following sources: National Institute of Health grant ALS-ENABLE (P30 GM124169), National Cancer Institute grant SBDR (CA92584), Department of Energy through Basic Energy Science grant DE-AC02-05CH11231 and Biological and Environmental Research grant IDAT. Beamline 8.3.1 at the Advanced Light Source is operated by the University of California at San Francisco with generous grants from the National Institutes of Health (R01 GM124149 and P30 GM124169), Plexxikon Inc. and the Integrated Diffraction Analysis Technologies program of the US Department of Energy Office of Biological and Environmental Research. This work was done in part using resources of the Shared Equipment Authority at Rice University (https://research.rice.edu/sea/). Molecular graphics and analyses were performed with UCSF Chimera, developed by the Resource for Biocomputing, Visualization, and Informatics at the University of California, San Francisco, with support from NIH P41-GM103311.

## Author contributions

**Chris A Brosey**: Conceptualization; Data curation; Formal analysis; Supervision; Investigation; Visualization; Methodology; Writing—original draft; Project administration; Writing—review and editing. **Runze Shen**: Formal analysis; Validation; Investigation; Methodology; Writing—review and editing. **John A Tainer**: Conceptualization; Supervision; Funding acquisition; Writing—original draft; Writing—review and editing.

Source data underlying figure panels in this paper may have individual authorship assigned. Where available, figure panel/source data authorship is listed in the following database record: biostudies:S-SCDT-10_1038-S44318-024-00360-6.

## Disclosure and competing interests statement

The authors declare no competing interests.

# Expanded View Figures

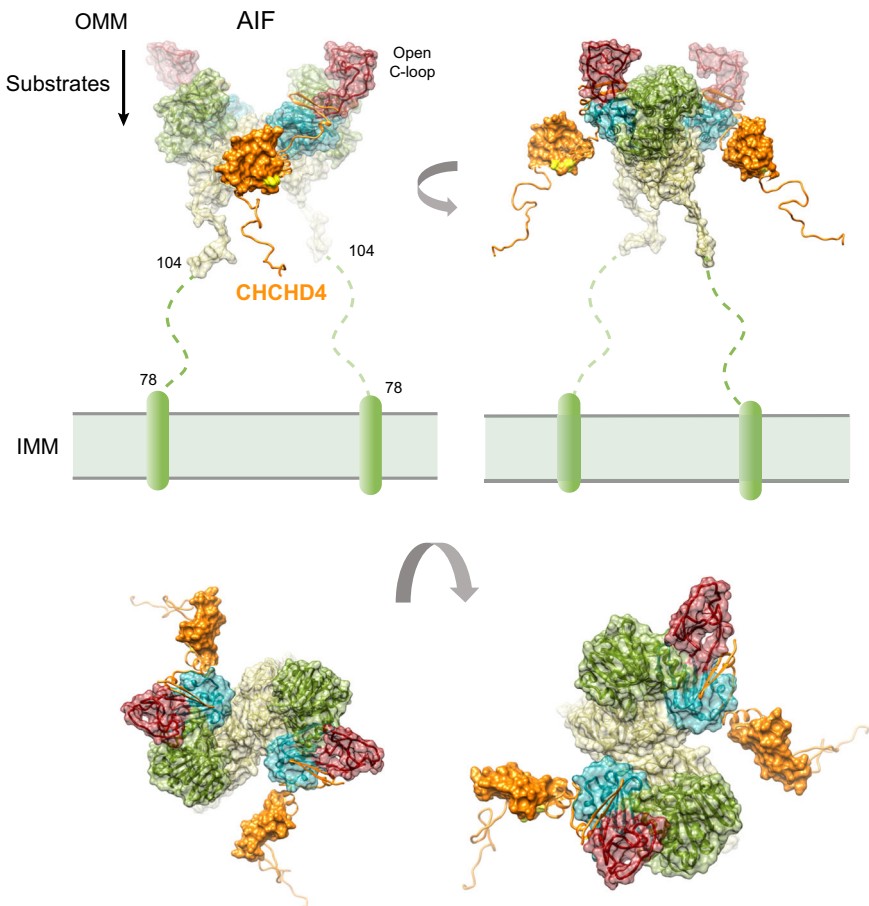

**Figure EV1.    Model of an AIF-CHCHD4 chaperone complex.**

The crystal structure of the AIF-W196A-CHCHD4-N45 chimera and NMR structure of the catalytic CHCH domain (PDB: 2K3J) were used as templates to generate a model of the full-length complex by building flexible segments with the Modeller extension of Chimera. AIF residues 78–104 were left unmodeled for clarity. Top panels exhibit side views relative to the mitochondrial membranes, while lower panels view the complex overhead from the OMM.

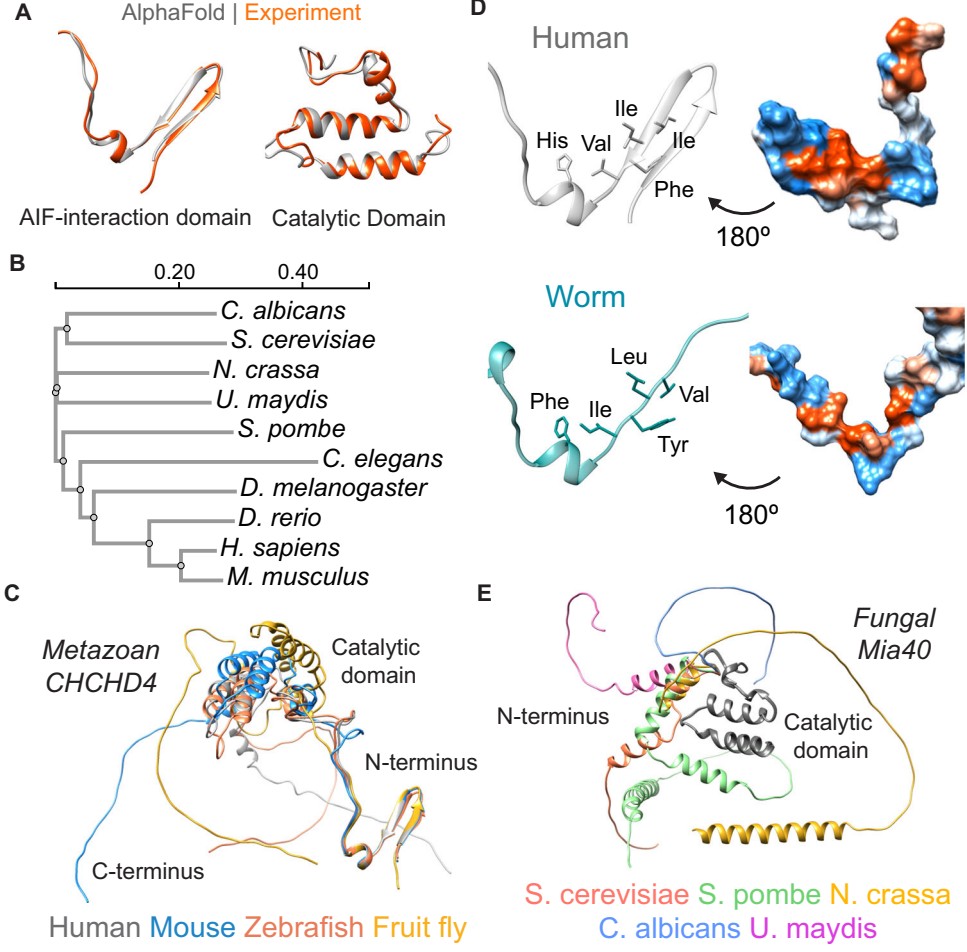

**Figure EV2. CHCHD4's AIF-interaction motif is conserved across metazoan CHCHD4 homologs.**

(A) Comparison of CHCHD4 experimental structures (orange) and AlphaFold models (gray) for the N-terminal AIF-interaction domain and central catalytic CHCH domain (PDB: 2K3J). (B) CHCHD4/Mia40 phylogenetic tree. Units indicate the average number of amino acid substitutions per site. (C) Superimposed AlphaFold models of metazoan CHCHD4 homologs. (D) The hydrophobic human AIF-interaction motif is conserved in *C. elegans* CHCHD4. Surfaces on right display Kyle–Doolittle hydrophobicity. (E) Superimposed AlphaFold models of fungal Mia40 homologs. C-terminal regions are excluded for clarity. Full-length models are displayed in Appendix Fig. S4.

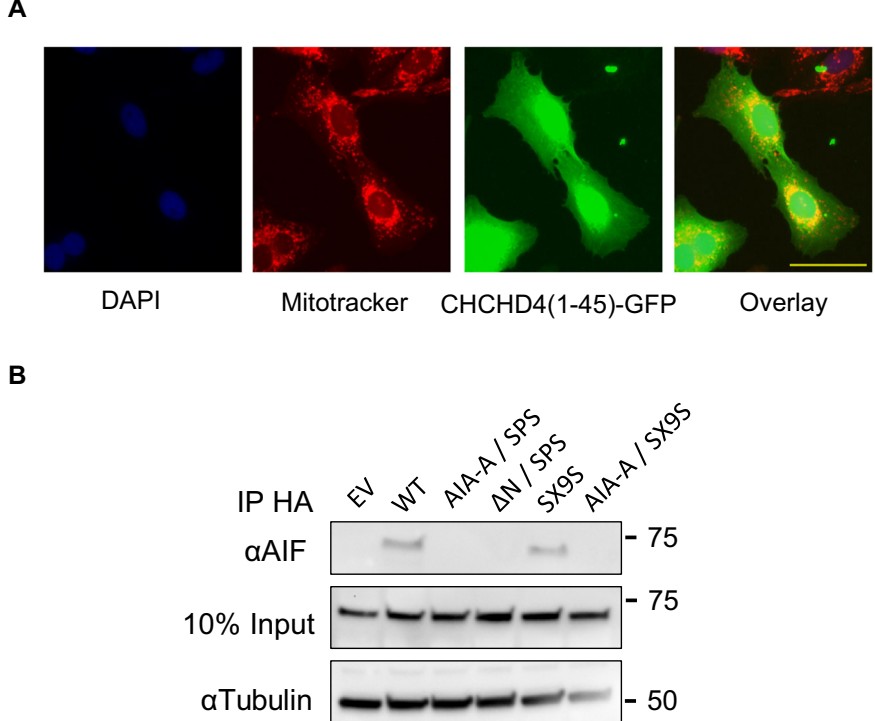

**Figure EV3.  CHCHD4's AIF-interaction domain is not an intrinsic mitochondrial localization sequence.**

(**A**) CHCHD4 (1–45) fused to a C-terminal GFP tag was transiently expressed in U2-OS cells, which were co-labeled with Hoescht (DAPI) and Mitotracker dyes for visualization. Results are representative of three independent experiments. The scale bar is 50 microns. (**B**) Endogenous AIF is immunoprecipitated from 293T whole-cell lysates by HA-tagged CHCHD4 mutants containing wild-type AIF-interaction motifs. Results are representative of three independent experiments. Source data are available online for this figure.

## A

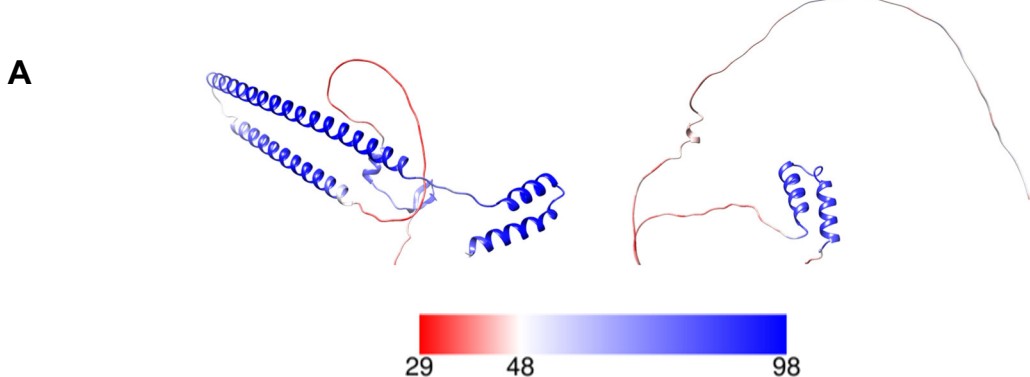

| **FL-SPS** | ΔH (kcal/mol) | -TΔS (kcal/mol) | ΔG (kcal/mol) | Kd (µM) | N |
|---|---|---|---|---|---|
| CHCHD6 | -5.77 (1.0) | -1.06 (1.17) | -6.13 (0.16) | 33.2 (9.4) | 1.17 (0.01) |
| CHCHD10 | -11.2 (0.26) | 4.83 (0.31) | -6.33 (0.18) | 23.5 (7.5) | 1.31 (0.12) |

| **ΔN-SPS** | ΔH (kcal/mol) | -TΔS (kcal/mol) | ΔG (kcal/mol) | Kd (µM) | N |
|---|---|---|---|---|---|
| CHCHD6 | -9.32 (0.28) | 2.76 (0.25) | 6.57 (0.04) | 15.5 (0.9) | 1.00 (0.04) |
| CHCHD10 | -18.63 (0.74) | 11.57 (0.78) | -7.07 (0.02) | 6.6 (0.2) | 1.11 (0.05) |

| **AIAA-SPS** | ΔH (kcal/mol) | -TΔS (kcal/mol) | ΔG (kcal/mol) | Kd (µM) | N |
|---|---|---|---|---|---|
| CHCHD6 | -4.78 (0.46) | -2.11 (0.40) | -6.89 (0.07) | 9.0 (1.0) | 1.16 (0.09) |
| CHCHD10 | -15.17 (0.67) | 7.86 (0.65) | -7.28 (0.04) | 4.6 (0.3) | 1.06 (0.06) |

**B**

**Figure EV4.   The AIF-interaction motif regulates CHCHD4 substrate binding.**

(A) AlphaFold models colored by pLDDT demonstrate high confidence in the helical hairpin structural prediction. (B) Summary of measured thermodynamic parameters from CHCHD4-substrate interactions. Values represent averages (standard deviations) from 2 to 3 independent experiments.

