## [Peer Review File · The EMBO Journal]

NADH-bound AIF activates the mitochondrial CHCHD4/MIA40 chaperone by a substrate-mimicry mechanism

John Tainer, Chris Brosey, and Runze Shen

Corresponding author(s): John Tainer (jtainer@mdanderson.org)

Review Timeline:

Submission Date:	30th Jun 24
Editorial Decision:	9th Aug 24
Revision Received:	10th Dec 24
Editorial Decision:	16th Dec 24
Revision Received:	19th Dec 24
Accepted:	20th Dec 24

Editor: Hartmut Vodermaier

Transaction Report:

Prof. John A. Tainer
MD Anderson Cancer Center
Department of Molecular and Cellular Oncology
University of Texas
Holcombe Blvd
Houston, TX 77030

9th Aug 2024

Re: EMBOJ-2024-118354

NADH-activated AIF leverages a substrate mimicry mechanism to stimulate mitochondrial CHCHD4/MIA40 chaperone activity

Dear Dr. Tainer,

Thank you for submitting your manuscript on AIF and CHCHD4 substrate mimicry to The EMBO Journal. I apologize for the considerable delay in getting back to you with a decision - I sent your study to three expert referees, but have so far still only received the reports of two of them. Since both of them are in fair agreement, and in the interest of time, I am now forwarding you these reports together with an invitation to start revising the study based on these overall supportive comments. As you will see, many points are focused on presentational aspects, but the two referees also raise a few experimental/technical questions that would be important to answer

Please note that this remains at present still a preliminary decision, and that any specific concerns raised by the outstanding third report (should it arrive within the next two weeks) may still need to be incorporated before finalization and resubmission of the work. I should also point out that our policy to allow only a single round of (major) revision makes it important to carefully respond to all points raised at this point. As always, competing work appearing during the course of the revision period will not affect our final decision on your study. Should you have any further questions linked to this decision, the referee report or the revision guidelines, please do not hesitate to contact me.

Thank you again for the opportunity to consider this work for The EMBO Journal, and I shall be in contact shortly to send you the outstanding third report, or to finalize the decision without it.

Yours sincerely,

Hartmut Vodermaier

9) To facilitate reproducibility and cross-laboratory adoption of methodologies, please structure the Materials & Methods section as outlined in our guide to authors, including a completed Reagents and Tools Table that can be downloaded from our author guidelines as well (<https://www.embopress.org/page/journal/14602075/authorguide#structuredmethods>).

10) Digital image enhancement is acceptable practice, as long as it accurately represents the original data and conforms to community standards. If a figure has been subjected to significant electronic manipulation, this must be clearly noted in the figure legend and/or the 'Materials and Methods' section. The editors reserve the right to request original versions of figures and the original images that were used to assemble the figure. Finally, we generally encourage uploading of numerical as well as gel/blot image source data; for details see: embopress.org/page/journal/14602075/authorguide#sourcedata

At EMBO Press, we ask authors to provide source data for the main manuscript figures. Our source data coordinator will contact you to discuss which figure panels we would need source data for and will also provide you with helpful tips on how to upload and organize the files.

In the interest of ensuring the conceptual advance provided by the work, we recommend submitting a revision within 3 months (7th Nov 2024). Please discuss the revision progress ahead of this time with the editor if you require more time to complete the revisions. Use the link below to submit your revision:

Link Not Available

Referee #2:

The manuscript by Brosey, Shen, & Tainer describes a tour-de-force structural and biophysical analysis of protein-protein interactions between CHCHD4 (also known as MIA40) and Apoptosis-Inducing Factor (AIF), an interaction that apparently plays a role in the mitochondrial oxidation-dependent protein import system. The manuscript is comprehensive, providing much data and several different kinds of data, including a crystal structure, SAXS, NMR, modeling, mitochondrial localization, and more. The volume and variety of the data is impressive. I reviewed only the X-ray crystallography and SAXS in any kind of detail. I ignored the mito import experiments and NMR, and my assessment of the sequence/Alphafold analysis, many gels, and SEC traces was cursory at best. Nevertheless, my impression is that this work has a very high degree-of-difficulty and represents a major contribution to the fields of AIF biochemistry and mitochondrial protein import.

I have a few minor comments/suggestions.

To aid crystallization of a complex, the authors cleverly devised a workaround by fusing the N-terminal 45 residues of CHCHD4 to AIF (so-called AIF-CHCHD4 chimera). In high altitude structural biology, one does what one must do. However, the conclusions of the work are subject to caveats related to the artificial nature of the system. The authors could admit/discuss this in the Discussion.

Although the PDB validation file is consistent with a high-quality crystal structure with good resolution (2.3 Å) and reveals no red flags, the authors should provide more information about the quality of the crystal structure, since it is the centerpiece of the work. It would be appropriate to provide electron density evidence for the bound N45 and list the average B-factors of AIF and N45 separately.

Figure S2B is confusing. The black and blue curves SAXS curves look almost identical but have very different R_g values of 26 and 38. And is there another curve "hiding" under the orange curve? Please improve the plot and its explanation.

The authors report SAXS data for AIF121, AIF121-NADH, and Chimera. Why didn't they compare these data to theoretical SAXS curves calculated from models? In particular, I should think that they would want to compare the SAXS data for the Chimera to the curve simulated from the crystal structure.

p. 8, CA RMSD of 0.938 seems optimistic given the uncertainties of 2.3 Å crystal structures. Perhaps 0.94 is sufficient. Similarly, consider changing 1.75 Å RMSD to 1.7 or 1.8.

The authors deposited their SAXS data in the SIMPLE SCATTERING database but not in the more well-known SASBDB. The authors should also deposit their SAXS data in SASBDB or tell us why we don't have to do that so I will know for my next manuscript.

Referee #3:

The mitochondrial disulfide relay mediates the import of proteins into and the oxidation within the intermembrane space. In yeast, electrons are transferred from substrates (i.e. newly imported proteins) via Mia40 and Erv1 to cytochrome c. Mia40 and Erv1 are conserved and, in animals, were re-named as CHCHD4 and ALR. The organization of the relay is more complex in animals, as CHCHD4 can bind to the NADH dehydrogenase AIF. AIF binding depends on the presence of NADH and stimulates CHCHD4-induced protein oxidation. However, the details of the interplay of AIF, CHCHD4, ALR and the substrates are not clear, despite some recently published structures of the AIF-CHCHD4 interaction.

This study clearly advances our understanding of the role of AIF in CHCHD4 activation. The authors show here for the first time that the N-terminal AIF-binding domain of CHCHD4 shows similarity to CHCHD4 substrates and might serve as a competitive inhibitor of the CHCHD4 substrate-binding pocket. The binding to AIF sequesters the N-terminal domain and therefore stimulates the chaperone and oxidoreductase activity of CHCHD4. This is an interesting observation of general relevance. This study is of high quality. However, the text and figures sometimes get lost in the details, which might restrict the readership to specialists of the mitochondrial thiol redox community. Thus, the authors should try to improve the figures and text to make it more accessible. The novelty and scientific relevance of this study certainly merits publication in a high-profile journal.

Specific points:

1. A major conclusion of the study is the 'substrate mimicry' of the CHCHD4-AIF interaction, which resembles that of the interactions between the CHCHD4/ALR and the CHCHD4/substrate dimers. The authors present some evidence by NMR shifts and an in vitro substrate oxidation assay, however, the data are difficult to interpret. In previous studies the substrate binding activity to CHCHD4 and ALR was measured by oxygen consumption assays which allow the quantitative assessment of the electron flux through the relay. The authors should use such a setup with and without the N45 fragment to show that it prevents substrate binding to CHCHD4.
2. The authors speculate the N-terminal domain of CHCHD4 serves as negative inhibitor which needs to be sequestered by AIF binding. Thus, a N-terminally truncated CHCHD4 version should show increased activity and function in an AIF-independent fashion. Is this really the case?
3. The PEG-MEM experiments are difficult to interpret as the authors test for the accessibility for cysteine residues rather than for the inaccessibility. Thus, no activity causes a ladder of bands owing to the inefficient alkylation with the bulky PEG-MEM probe. The authors therefore should better use inverted shift assays for which samples are first treated with NEM and then reduced with TCEP before the initially oxidized thiols are alkylated with PEG-MEM. Thereby, CHCHD4 activity will lead to shifts to the fully oxidized form. In order to modify all reduced thiols without steric hindrance, smaller reagents such as AMS might be better suited.
4. The model in Fig. 6 is not very informative as it shows proteins as blobs without molecular resolution. The authors should better show the role of AIF binding for the two domains of CHCHD4 with higher resolution in order to summarize the major finding of this study. This will be more valuable for readers even if some parts of the CHCHD4 structure remain unknown (or are generally unstructured). The current figure 4 should better be moved to the supplement.
5. The structure of CHCHD4 from residue 45 to 105 was already known. The authors here show for the first time the structure of residues 1 to 28 of CHCHD4, bound to AIF. The authors should use these two structures and come up with a model of how rest of CHCHD4, including its substrate-binding domain, is positioned in the AIF-bound complex.
6. Page 6: CHCHD4 is misspelled as CHCDH4

NADH-bound AIF activates the mitochondrial CHCHD4/MIA40 chaperone by a substrate mimicry mechanism

Reviewer Responses

We would like to thank the reviewers for their careful and thorough review of our study and the time required to provide constructive feedback. We believe addressing these recommendations has greatly improved the quality, rigor, and clarity of the work.

In addition to addressing specific concerns, we have given special attention to improving the clarity of the manuscript figures and layout. To improve the organization and visual presentation of the data, we have reorganized the existing figures (table below), in addition to incorporating new data requested by the reviewers. These changes include the following:

- Figure 5 has been expanded into Figures 4-6. Diagrams describing the associated experiments are now also included.
- Figure 6 has been revised as Figure 7 to illustrate a sequenced progression of CHCHD4, AIF, and ALR interactions in a mechanistic model of disulfide relay import.
- Five expanded view (EV) figures and tables have been assigned from the main and appendix figures.
- Appendix Figure 3 (SAXS modeling) and Table 3 (CHCHD4-N45 B-factors) have been added to the Appendix.
- Figure S8 has been divided into Appendix Figures 6-7 to include supplementary data associated with the Amplex Red fluorescent hydrogen peroxide detection assay.

Original Manuscript	Revised Manuscript	Original Manuscript	Revised Manuscript
Figure 1	Figure 1	Figure S1	Appendix Figure 1
Figure 2	Figure EV2	Figure S2	Appendix Figure 2
Figure 3	Figure 2	Figure S3	Figure EV1
Figure 4	Figure 3	Figure S4	Appendix Figure 4
Figure 5	Figures 4-6	Figure S5	Figure EV3
Figure 6	Figure 7	Figure S6	Appendix Figure 5
		Figure S7	Figure 4 / Figure EV4
Table 1	Table 1	Figure S8	Appendix Figure 6-7
Table 2	Table EV1	Figure S9	Appendix Figure 8
		Figure S10	Appendix Source Data
		Table S1	Appendix Table 1
		Table S2	Appendix Table 2

We have also corrected the following:

- A mislabeling of ITC thermograms for CHCHD10 in Figure 4E (AIA-A and ΔN switched).
- The statistical t-tests reported in Figures 5E and 6F were updated to include Welch's correction to account for unequal variances between conditions.

All other changes to the text are indicated in yellow highlight.

Referee #2:

The manuscript by Brosey, Shen, & Tainer describes a tour-de-force structural and biophysical analysis of protein-protein interactions between CHCHD4 (also known as MIA40) and Apoptosis-Inducing Factor (AIF), an interaction that apparently plays a role in the mitochondrial oxidation-dependent protein import system. The manuscript is comprehensive, providing much data and several different kinds of data, including a crystal structure, SAXS, NMR, modeling, mitochondrial localization, and more. The volume and variety of the data is impressive. I reviewed only the X-ray crystallography and SAXS in any kind of detail. I ignored the mito import experiments and NMR, and my assessment of the sequence/AlphaFold analysis, many gels, and SEC traces was cursory at best. Nevertheless, my impression is that this work has a very high degree-of-difficulty and represents a major contribution to the fields of AIF biochemistry and mitochondrial protein import.

I have a few minor comments/suggestions.

To aid crystallization of a complex, the authors cleverly devised a workaround by fusing the N-terminal 45 residues of CHCHD4 to AIF (so-called AIF-CHCHD4 chimera). In high altitude structural biology, one does what one must do. However, the conclusions of the work are subject to caveats related to the artificial nature of the system. The authors could admit/discuss this in the Discussion.

The following has been added to the Discussion:

'Though the AIF-CHCHD4 chimeric fusion used in structure determination bears the caveats of an engineered system, biochemical and cellular testing robustly support the revealed binding interface and its functional relevance for CHCHD4 chaperone activity.'

We link this comment to discussion of the low-resolution structure of the mouse CHCHD4 peptide that supports a similar binding arrangement of the AIF-CHCHD4 interface.

Although the PDB validation file is consistent with a high-quality crystal structure with good resolution (2.3 Å) and reveals no red flags, the authors should provide more information about the quality of the crystal structure, since it is the centerpiece of the work. It would be appropriate to provide electron density evidence for the bound N45 and list the average B-factors of AIF and N45 separately.

We agree. We now include figure panels displaying CHCHD4 electron density (Appendix Figure 2D) and a supplementary table providing average and median B-factor values for each AIF and CHCHD4-N45 molecule within the crystal structure (Appendix Table 3).

Figure S2B is confusing. The black and blue curves SAXS curves look almost identical but have very different R_g values of 26 and 38. And is there another curve "hiding" under the orange curve? Please improve the plot and its explanation.

The blue AIF-NADH dimer $I(q)$ curve was indeed 'hidden' by the AIF-CHCHD4-chimera $I(q)$ curve – which underscores the high similarity between the native AIF dimer and the chimeric construct. We have reordered the position of the curves in the $I(q)$ plot and increased the line thickness in both $I(q)$ and normalized Kratky plots to highlight their differences more effectively. We have also included a zoomed inset of the low- q region of the $I(q)$ to display the overlay in more detail.

The authors report SAXS data for AIF121, AIF121-NADH, and Chimera. Why didn't they compare these data to theoretical SAXS curves calculated from models? In particular, I should think that they would want to compare the SAXS data for the Chimera to the curve simulated from the crystal structure.

The AIF121 and AIF121-NADH curves were originally characterized in Brosey et al, *Structure* (2016) (DOI: 10.1016/j.str.2016.09.012) and aligned well to theoretical scattering curves from monomeric and dimeric AIF crystal structures. Comparing similarity between the AIF121 and Chimera SAXS data is a useful comparison, and we now include this in Appendix Figure 2B.

We have also added a BilboMD analysis and minimal ensemble search comparing the AIF-CHCHD4-chimera SAXS data to a model of the chimera based upon the crystal structure (Appendix Figure 3) and include the following in the results section:

To compare the crystal structure to the solution state observed by small-angle X-ray scattering, we added residues missing from the crystal structure and submitted the resulting model to BilboMD simulation and minimal ensemble search (MES). BilboMD generates a population of models via constrained molecular dynamics and identifies the minimum number of models that best describe the data. For the AIF-CHCHD4 chimera, the MES algorithm returned a two-model ensemble (Appendix Figure 3A-B), supporting flexibility at the N- and C-termini and C-loop. Notably, one of the CHCHD4-N45 fragments of the minor population is captured in an unbound state, suggesting an equilibrium between bound and unengaged CHCHD4-N45 in solution that favors the AIF-CHCHD4-N45 complex.

p. 8, CA RMSD of 0.938 seems optimistic given the uncertainties of 2.3 Å crystal structures. Perhaps 0.94 is sufficient. Similarly, consider changing 1.75 Å RMSD to 1.7 or 1.8.

We have changed the indicated $C\alpha$ RMSD values of 0.938 to 0.94 Å and 1.75 to 1.8 Å.

The authors deposited their SAXS data in the SIMPLE SCATTERING database but not in the more well-known SASBDB. The authors should also deposit their SAXS data in SASBDB or tell us why we don't have to do that so I will know for my next manuscript.

Yes, this is an important clarification. The SIMPLE SCATTERING data base curates correlated sets of experimental scattering curves (Murray et al, *Meth Enzymol*, 2023, DOI: 10.1016/bs.mie.2022.09.024). This includes SAXS data collected for time-resolved series, buffer or ligand screens, or panels of mutants. We originally chose this database for deposition, since the CHCHD4 datasets were collected as a mutant panel meant for internal comparison (rather than direct shape analysis). The SIMPLE SCATTERING database also contains previously published high-throughput, time-resolved data from monomeric and NADH-loaded dimeric AIF, and we thought it would be helpful to make the AIF-CHCHD4 chimeric data (also collected in a similar format) accessible in this venue.

The point raised about broad accessibility to the community is an important one, so we have also deposited the scattering data in SASBDB. The entries can be previewed at the following URLs:

Dataset	SASBDB Entry
AIF-CHCHD4 chimera	https://www.sasbdb.org/data/SASDV47/41qhhldk3u
CHCHD4 wild-type	https://www.sasbdb.org/data/SASDVY6/yeazxamw7p
CHCHD4 F12A/I14A/H20A, 'AIA-A'	https://www.sasbdb.org/data/SASDVZ6/2yov2uwpgx
CHCHD4 H20D	https://www.sasbdb.org/data/SASDV27/0p6slh9cv1
CHCHD4 L28D	https://www.sasbdb.org/data/SASDV37/bmxjon3n1r

Referee #3:

The mitochondrial disulfide relay mediates the import of proteins into and the oxidation within the intermembrane space. In yeast, electrons are transferred from substrates (i.e. newly imported proteins) via Mia40 and Erv1 to cytochrome c. Mia40 and Erv1 are conserved and, in animals, were re-named as CHCHD4 and ALR. The organization of the relay is more complex in animals, as CHCHD4 can bind to the NADH dehydrogenase AIF. AIF binding depends on the presence of NADH and stimulates CHCHD4-induced protein oxidation. However, the details of the interplay of AIF, CHCHD4, ALR and the substrates are not clear, despite some recently published structures of the AIF-CHCHD4 interaction.

This study clearly advances our understanding of the role of AIF in CHCHD4 activation. The authors show here for the first time that the N-terminal AIF-binding domain of CHCHD4 shows similarity to CHCHD4 substrates and might serve as a competitive inhibitor of the CHCHD4 substrate-binding pocket. The binding to AIF sequesters the N-terminal domain and therefore stimulates the chaperone and oxidoreductase activity of CHCHD4. This is an interesting observation of general relevance. This study is of high quality. However, the text and figures sometimes get lost in the details, which might restrict the readership to specialists of the mitochondrial thiol redox community. Thus, the authors should try to improve the figures and text to make it more accessible. The novelty and scientific relevance of this study certainly merits publication in a high-profile journal.

Specific points:

1. A major conclusion of the study is the 'substrate mimicry' of the CHCHD4-AIF interaction, which resembles that of the interactions between the CHCHD4/ALR and the CHCHD4/substrate dimers. The authors present some evidence by NMR shifts and an *in vitro* substrate oxidation assay, however, the data are difficult to interpret.

To increase clarity of NMR, ITC, and *in vitro* chaperone experimental design, we have added diagrams of the CHCHD4 constructs/mutants under consideration, as well as diagrams of each experiment where appropriate. We have also reorganized the figures to divide Figure 5 into 3 separate figures (Figure 4-6) to provide better visual flow and clarity to the data.

In previous studies the substrate binding activity to CHCHD4 and ALR was measured by oxygen consumption assays which allow the quantitative assessment of the electron flux through the relay. The authors should use such a setup with and without the N45 fragment to show that it prevents substrate binding to CHCHD4.

This is a good suggestion. Monitoring molecular reactants/products of the disulfide relay reactions is an insightful complementary assay to the CHCHD6/CHCHD4 chaperone experiments which detect substrate disulfide formation. We have chosen to use Amplex Red dye to monitor hydrogen peroxide production arising from transfer of electrons from ALR to solvent oxygen as the final electron acceptor. This assay format was previously used to monitor oxidation of Tim13 by the yeast Mia40-Erv1 system (DOI: 10.1091/mbc.E08-10-1062). Fluorescent dyes for monitoring oxygen depletion are available; however, we selected Amplex Red peroxide detection for its sensitivity and robustness, its routine application to a variety of *in vitro* enzymatic assays, and its precedence in the literature. Our control time-resolved experiments with wild-type CHCHD4, unfolded CHCHD6 substrate, and ALR demonstrate a robust time-dependent increase in peroxide production similar to that reported for the yeast system (Figure 5G). Moreover, they show an increased rate of peroxide generation from the CHCHD4-AIA-A mutant, consistent with its enhanced chaperone activity. These results are now incorporated as Figure 5F-H.

Titration of the N45 fragment *in trans* presents many practical challenges, since the free fragment must compete with an equivalent fragment that is covalently linked and hydrophobically associated with its intramolecular binding partner (CHCHD4's central catalytic domain). At the same time, the exposed hydrophobic binding surface of the free N45 fragment would be expected to favor its self-association, hindering the availability of free N45 fragment at the higher concentrations required to compete against the local concentration effects of a linked fragment.

However, we moved forward with testing N45 titration in the Amplex Red assay. During these investigations, we discovered that the fragment produced its own high fluorescent background signal, which we attributed to the free cysteine thiol at residue 4. We subsequently treated the wild-type N45 domain with maleimide to shield this cysteine and also prepared an N45-C4S mutant to remove the thiol. These modifications addressed the background, allowing for titration of the wild-type and mutant fragments to concentrations of 50-fold molar excess relative to full-length wild-type and mutant (AIA-A) CHCHD4. Even at these concentrations, we did not observe a reduction in peroxide production, consistent with the expectation that the competition format does not optimally probe the interaction.

Our reported point mutations at CHCHD4's intramolecular binding interface (AIA-A, H20D) demonstrate structural (Figure 3) and functional (Figure 4-5) changes consistent with altering CHCHD4 intramolecular contact, supporting the proposed model for intramolecular regulation of CHCHD4.

2. The authors speculate the N-terminal domain of CHCHD4 serves as negative inhibitor which needs to be sequestered by AIF binding. Thus, a N-terminally truncated CHCHD4 version should show increased activity and function in an AIF-independent fashion. Is this really the case?

This is a great point, which we discuss in the results section dedicated to CHCHD4 substrate interactions and chaperone activity (**The AIF-interaction motif regulates CHCHD4 substrate binding and chaperone activity**). The ITC experiments on N-terminally truncated CHCHD4 (with redox-inactive SPS motif) do indeed demonstrate *increased* affinity for unfolded CHCHD6 and CHCHD10 substrates (Figures 4, EV4). Thus, CHCHD4 substrate binding is enhanced by removing the N-terminal domain.

However, removing the N-terminal domain from redox-active CHCHD4 does not result in enhanced chaperone activity in the *in vitro* refolding assay, as one would expect ('Notably, removing the N-terminus does not enhance CHCHD4 chaperone activity (Fig. 5E)'). On the contrary, its marginal activity resembles the inactive SPS mutant. We proposed that the truncated mutant exhibits hindered activity because of self-association through the now-exposed hydrophobic substrate binding site, resulting in formation of bridging intermolecular disulfides between proximate CPC motifs. This 'disulfide dimer' between two truncated mutants would be expected to prevent engagement with CHCHD4 substrates.

We provide evidence for such a disulfide-mediated dimer formed by the redox-active truncation mutant using analytical size-exclusion chromatography and non-reducing SDS-PAGE analysis (Appendix Figure 7). The persistence of a portion of the Δ N-CHCHD4 dimer at high levels of reducing agent underscores the difficulty of monomer separation once the intermolecular disulfide has formed and supports the importance of the N-terminal domain in shielding the CHCHD4 active site to prevent adventitious binding and reaction with the CPC motif.

CHCHD4 mutants that alter but do not eliminate the intramolecular interaction between the N-terminal domain and catalytic central domain exhibit increased chaperone and peroxide activity (AIA-A, H20D, Fig. 5E, 5H), as well as substrate binding (AIA-A, Fig. 4E-F), while still protecting the redox-sensitive CPC motif.

We have tried to express this more clearly in the text (in highlights below) and by including diagrams of the disulfide-mediated Δ N-CHCHD4 dimer in Appendix Figure 7.

Notably, removing the N-terminus does not enhance CHCHD4 chaperone activity. In fact, the truncation mutant exhibits the marginal activity of the inactive SPS mutant (Fig. 5E). Unlike Δ N-CHCHD4-SPS samples used in the NMR and ITC experiments, this construct retains a redox-sensitive CPC motif. Since the refolding assay is performed in the absence of reducing agent,

we considered whether ΔN -CHCHD4 molecules form or retain inactivating intermolecular disulfide bonds, which would prevent successful engagement with substrates.

3. The PEG-MEM experiments are difficult to interpret as the authors test for the accessibility for cysteine residues rather than for the inaccessibility. Thus, no activity causes a ladder of bands owing to the inefficient alkylation with the bulky PEG-MEM probe. The authors therefore should better use inverted shift assays for which samples are first treated with NEM and then reduced with TCEP before the initially oxidized thiols are alkylated with PEG-MEM. Thereby, CHCHD4 activity will lead to shifts to the fully oxidized form. In order to modify all reduced thiols without steric hindrance, smaller reagents such as AMS might be better suited.

We appreciate the reviewer's point regarding the bulkiness of the PEG-MEM cysteine probe. The choice to use PEG-MEM involved two factors. First, we were interested in resolving and monitoring the oxidation status of all 4 cysteines in the CHCHD6 substrate, particularly those contributing to the helical hairpin's interior disulfide that is chaperoned by CHCHD4 and formed first. The 10-kDa PEG-MEM allows unambiguous resolution of substrates containing 1, 2, 3, or 4 free thiols by SDS-PAGE. Disappearance of the 3- and 4-thiol species points to chaperoned formation of the interior disulfide. Using the smaller 0.5-kDa AMS label in place of PEG-MEM would make it difficult to resolve and visualize these individual species by SDS-PAGE.

Second, we wanted a direct label to detect the state of the system with a minimum lag time. One concern with treating with a sequence of NEM, TCEP, and PEG-MEM is the potential for competitive interference between NEM and PEG-MEM in labeling newly TCEP-exposed thiols. In cellular experiments that use the inverse labeling approach (DOI: 10.15252/embj.2019103889), wash and lysis steps separate *in vivo* NEM labeling from *in vitro* TCEP and PEG-MEM treatment. For an *in vitro* refolding experiment with purified proteins, achieving separation of NEM and TCEP/PEG-MEM treatments becomes complicated without complex sample manipulations. Introducing TCEP into the refolding reactions also risks introducing background from TCEP-reduced and PEG-MEM modified CHCHD4 and ALR thiols.

While the bulkiness of the PEG-MEM label may hinder complete modification of all cysteines within the CHCHD6 substrate, the comparative design and controls of the experiment should still allow for assessment of the enhanced activity of the activated CHCHD4 mutants. In this environment, the assay effectively ranks activity across absent or catalytically inactive CHCHD4, wild-type CHCHD4, and the activated CHCHD4 mutants (Figure 5).

4. The model in Fig. 6 is not very informative as it shows proteins as blobs without molecular resolution. The authors should better show the role of AIF binding for the two domains of CHCHD4 with higher resolution in order to summarize the major finding of this study. This will be more valuable for readers even if some parts of the CHCHD4 structure remain unknown (or are generally unstructured). The current figure 4 should better be moved to the supplement.

We agree that emphasizing the mechanistic details of our proposed model for CHCHD4 regulation would make for a stronger figure. To this end, we have reworked the final figure (now Figure 7) to show a sequenced progression of CHCHD4, AIF, and ALR interactions during the disulfide relay under conditions of abundant and limiting NADH.

When referencing Figure 4, we assume the reviewer is referring to the SAXS and NMR data. We respectfully disagree with the suggestion to move this figure to the supplement, as the NMR data provides key support for the presence of inter-domain contact between the CHCHD4 N-terminal and central domains. We have now added diagrams of the CHCHD4 mutants to emphasize the structural interactions revealed by this data (Figure 3B).

5. The structure of CHCHD4 from residue 45 to 105 was already known. The authors here show for the first time the structure of residues 1 to 28 of CHCHD4, bound to AIF. The authors should use these two structures and come up with a model of how rest of CHCHD4, including its substrate-binding domain, is positioned in the AIF-bound complex.

A good suggestion – a model for full-length CHCHD4 bound to dimeric NADH-activated AIF was originally constructed as suggested from the X-ray structure (CHCHD4 residues 1-30) and the NMR structure (CHCHD4 residues 45-109). This was displayed in Figure S3 and discussed at the end of the results section focused on the X-ray structure. The right panel of the original Figure 6 also contains a version of this model. To provide greater emphasis to the model, we now display it in Expanded View Figure 1. As part of reworking Figure 6 to create Figure 7, we have also emphasized the IMM-placement of CHCHD4 and the ALR regeneration hand-off mechanism suggested by the model.

To emphasize the integration of X-ray and NMR data in constructing the model, we have also added following to the text:

Leveraging our X-ray structure and the published NMR structure of the human CHCHD4 catalytic domain (PDB: 2K3J) (Banci et al., 2009), we constructed a model of full-length CHCHD4 flexibly extending from the AIF-anchored N-terminal domain (Fig. EV1).

6. Page 6: CHCHD4 is misspelled as CHCDH4

Thank you for catching this – it has been corrected.

Prof. John A. Tainer
MD Anderson Cancer Center
Department of Molecular and Cellular Oncology
University of Texas
Holcombe Blvd
Houston, TX 77030

16th Dec 2024

Re: EMBOJ-2024-118354R
NADH-bound AIF activates the mitochondrial CHCHD4/MIA40 chaperone by a substrate mimicry mechanism

Dear John,

Thank you again for submitting your revision to The EMBO Journal. I have now carefully checked your responses to the original comments as well as their incorporation into the revised manuscript, and found all points satisfactorily addressed. We shall therefore be happy to proceed with acceptance and publication of this work, as soon as the following remaining editorial points have been taken care of:

- Title page: please add a superscript label for the corresponding author; and in the title, please add a hyphen to clarify the compound modifier in "substrate-mimicry mechanism".
- Please double-check to make sure to all relevant funding information in the manuscript is congruent with the info entered into our submission system (Robert A. Welch Chemistry Chair (G-0010) appears to be missing in the submission system).
- Please correct the reference list, making sure that for references with more than 10 authors on a paper, only the first 10 should be listed, followed by 'et al.' (please refer to our Guide to Authors for additional information on EMBO J reference format).
- As we are switching from a free-text author contribution statement towards a more formal statement based on Contributor Role Taxonomy (CRediT) terms, please remove the present Author Contribution section and instead specify each author's contribution(s) directly in the Author Information page of our submission system during upload of the final manuscript. See <https://casrai.org/credit/> for more information.
- On the first page of the Appendix PDF file, please include the header "Appendix for" above the title line (you may remove the affiliations here instead), and in the Table of Contents, make sure to write "Appendix Figure/Table S..." instead of just "Figure/Table S...".
- Please add at least one call-out to Appendix Figure S8 within the text, which currently seems missing.
- If the table with SEC-MALS Analysis data should remain in the main manuscript (like the table with the MST affinity measurements), please rename it "Table 2" in its header and throughout the text. If it should instead remain as an expanded view table ("Table EV1"), please remove it from the main text and upload it as a separate DOCX or XLSX file.
- Please correct the legend to Figure 6, which has become mixed-up during the remaking of the figure arrangements: It seems that a description for panel D is missing, what is described as D may refer to E, what is described as E may be F, and "J" may refer again to E?
- Since some of the data in Figure 4G come from only 2 independent experiments, please change the presentation from bar diagram with average + SD into simply plotting the individual data points for each of the six conditions.
- Please define the center of the error bars in the legends of figures 4G, 5C, E, H.
- Please indicate the exact p values in the legends of figures 5E, H, 6F.
- Please provide suggestions for a short 'blurb' text prefacing and summing up the conceptual aspect of the study in two sentences (max. 250 characters), followed by 3-5 one-sentence 'bullet points' with brief factual statements of key results of the paper; they will form the basis of an editor-written 'Synopsis' accompanying the online version of the article. Please also upload a synopsis image, which can be used as a "visual title" for the synopsis section of your paper. The image (maybe based on a part of Figure 7?) should be in PNG or JPG format, and please make sure that it remains in the modest dimensions of (exactly) 550 pixels wide and 300-600 pixels high.

- Finally, please make sure to remove all text mark-up/highlighting at this stage of the process.

I am therefore returning the manuscript to you for a final round of minor revision, to allow you to make these modifications and upload the revised files. Once we will have received them, we should be ready to swiftly proceed with formal acceptance and production of the manuscript.

With kind regards,

Hartmut

9) To facilitate reproducibility and cross-laboratory adoption of methodologies, please structure the Materials & Methods section as outlined in our guide to authors, including a completed Reagents and Tools Table that can be downloaded from our author guidelines as well (<https://www.embopress.org/page/journal/14602075/authorguide#structuredmethods>).

10) Digital image enhancement is acceptable practice, as long as it accurately represents the original data and conforms to community standards. If a figure has been subjected to significant electronic manipulation, this must be clearly noted in the figure legend and/or the 'Materials and Methods' section. The editors reserve the right to request original versions of figures and the original images that were used to assemble the figure. Finally, we generally encourage uploading of numerical as well as gel/blot image source data; for details see: embopress.org/page/journal/14602075/authorguide#sourcedata

At EMBO Press, we ask authors to provide source data for the main manuscript figures. Our source data coordinator will contact you to discuss which figure panels we would need source data for and will also provide you with helpful tips on how to upload and organize the files.

Further information is available in our Guide For Authors:

In the interest of ensuring the conceptual advance provided by the work, we recommend submitting a revision within 3 months (16th Mar 2025). Please discuss the revision progress ahead of this time with the editor if you require more time to complete the revisions. Use the link below to submit your revision:

Link Not Available

Prof. John A. Tainer
MD Anderson Cancer Center
Department of Molecular and Cellular Oncology
University of Texas
Holcombe Blvd
Houston, TX 77030

20th Dec 2024

Re: EMBOJ-2024-118354R1
NADH-bound AIF activates the mitochondrial CHCHD4/MIA40 chaperone by a substrate-mimicry mechanism

Dear Prof. Tainer,

Thank you for submitting your final revised manuscript for our consideration. I am pleased to inform you that we have now accepted it for publication in The EMBO Journal.

Yours sincerely,

Hartmut Vodermaier
